# Novel metabolic role for BDNF in pancreatic β-cell insulin secretion

Gianluca Fulgenzi [1,4], Zhenyi Hong[1,4], Francesco Tomassoni-Ardori[1], Luiz F. Barella [2], Jodi Becker[1],
Colleen Barrick[1], Deborah Swing[1], Sudhirkumar Yanpallewar[1], Brad St Croix[1], Jürgen Wess[2], Oksana Gavrilova[3] &
Lino Tessarollo[1✉]

BDNF signaling in hypothalamic circuitries regulates mammalian food intake. However, whether BDNF exerts metabolic effects on peripheral organs is currently unknown. Here, we show that the BDNF receptor TrkB.T1 is expressed by pancreatic β-cells where it regulates insulin release. Mice lacking TrkB.T1 show impaired glucose tolerance and insulin secretion. β-cell BDNF-TrkB.T1 signaling triggers calcium release from intracellular stores, increasing glucose-induced insulin secretion. Additionally, BDNF is secreted by skeletal muscle and muscle-specific BDNF knockout phenocopies the β-cell TrkB.T1 deletion metabolic impairments. The finding that BDNF is also secreted by differentiated human muscle cells and induces insulin secretion in human islets via TrkB.T1 identifies a new regulatory function of BDNF on metabolism that is independent of CNS activity. Our data suggest that muscle-derived BDNF may be a key factor mediating increased glucose metabolism in response to exercise, with implications for the treatment of diabetes and related metabolic diseases.

[1] Mouse Cancer Genetics Program, CCR, NCI, NIH, Frederick, USA. [2] Molecular Signaling Section, Laboratory of Bioorganic Chemistry, NIDDK, NIH, Bethesda, USA. [3] Mouse Metabolism Core Laboratory, NIDDK, NIH, Bethesda, USA. [4] These authors contributed equally: Gianluca Fulgenzi, Zhenyi Hong. ✉email: tessarol@mail.nih.gov

nsulin is the body hormone capable of lowering blood glucose levels and impairments in its secretion or a reduction in body sensitivity to its function leads to diabetes. Pancreatic β-cells secrete insulin following depolarization caused by glucose uptake and metabolism. Extensive knowledge has been accumulated on the properties of ion channels regulating β-cell membrane electrical activity while the pathways modulating their function are still less defined[1]. Brain Derived Neurotrophic Factor (BDNF) is a potent pro-survival factor for neurons of the central and peripheral nervous system and a strong modulator of brain synaptic plasticity[2–4]. Signaling defects caused by dysregulation of its high affinity TrkB receptors have been linked to the pathophysiology of several neurological and neurodegenerative disorders[5]. In addition, mice and humans heterozygous for BDNF or TrkB inactivating mutations develop hyperphagic obesity[6–8] due to defects in hypothalamic anorexigenic and orexigenic signaling pathways[7,9]. Interestingly, systemic BDNF injections improve blood glucose levels and alleviate fasting hyperglycemia in mouse models of obesity[10] through a still unknown mechanism. The sources and mechanisms leading to changes in peripheral BDNF levels are still unknown. Several studies have reported exercise induced BDNF expression in skeletal muscle[11,12]. Because of the large body skeletal muscle mass, this organ has the potential to be a primary source of systemic BDNF. However, to date, there is no direct evidence that skeletal muscle can secrete BDNF: in mice most likely because of the technical difficulties in detecting low levels of BDNF[13]; in humans, because BDNF is stored in platelets from which it can be released during sampling, leading to confounding quantitative analyses[14–16].

In mammals there are two major isoforms of the BDNF receptor TrkB: a full-length receptor with an extracellular ligand binding domain and an intracellular kinase domain used for signaling (TrkB.FL); and a TrkB receptor lacking kinase activity (TrkB.T1) due to the absence of the tyrosine kinase domain. These receptors are produced by alternative splicing but both have a unique polyadenylation signal[4]. While TrkB.FL is expressed mainly in neurons of the central nervous system (CNS) and the peripheral nervous system (PNS), TrkB.T1 shows widespread expression outside of the nervous system in humans[17]. Of particular interest is the presence of polyadenylated truncated TrkB.T1 mRNA in the pancreas suggesting that in this organ expression of TrkB.T1 is independent of neuronal innervation. Here, we demonstrate that the BDNF-TrkB.T1 signaling in pancreas promotes glucose-induced insulin release. Moreover, skeletal muscle secretes BDNF providing a peripheral source of this neurotrophin and contributing to the regulation of glucose metabolism.

## Results

### Pancreatic β-cells express the BDNF receptor TrkB.T1. We used the mouse as a model organism to investigate the biological significance of pancreatic TrkB receptors. We first tested whether TrkB mRNA is translated and whether it is present in the exocrine or endocrine pancreas. Immunoprecipitation experiments with an antibody recognizing the TrkB extracellular domain, showed significant levels of a truncated TrkB receptor isoform. Although truncated TrkB is difficult to detect in unfractionated pancreatic tissue by immunoblotting, the identity of this receptor as TrkB.T1 was confirmed at the protein and mRNA level by immunoprecipitation and reverse-transcription PCR (RT-PCR), respectively, using pancreatic lysates from TrkB.T1+/+ or −/− mice (Fig. 1a and Supplementary Fig. 1). To investigate whether this receptor is expressed at low level throughout the pancreas or localized in the islets, which represent only a small fraction of the total organ mass, we isolated adult mouse islets for western

analysis. Figure 1b shows that TrkB.T1 could be readily detected in lysates of mouse islets by immunoblotting suggesting that BDNF may regulate pancreas endocrine functions. Importantly, RT-PCR analysis showed that TrkB.T1 is the most abundant Trk receptor isoform in islets since both TrkA and TrkC were present at negligible levels compared to TrkB.T1 (Supplementary Fig. 1). Our data is also in agreement with expression data from the human pancreatic islet transcriptome indicating that TRKB.T1 is the highest expressed Trk receptor isoform in human islets[18]. Because sensitive and specific antibodies for the different TrkB receptor isoforms are currently not available, we used CRISPR/Cas9 technology to knock-in (KI) a V5-epitope tag in the TrkB locus, at the C-terminus of the TrkB.T1 genomic isoform (TrkB. T1-V5; Fig. 1c and Supplementary Fig. 2) to unequivocally identify the expression pattern of this receptor. Supplementary Fig. 2A shows the specific tagging of the TrkB.T1 isoform in the TrkB.T1-V5 KI mouse brain. Importantly, the V5 KI mouse faithfully recapitulated the expression pattern of the endogenous TrkB.T1 protein in mouse brain (Supplementary Fig. 2A), validating the usefulness of this model for further expression profiling in other tissues with lower-level expression. Furthermore, immunofluorescence analysis of TrkBT1-V5 mouse pancreata using a V5-specific antibody, in association with the use of antibodies recognizing, insulin, glucagon, somatostatin or CD31, showed that TrkB.T1 was expressed exclusively in the insulin secreting islet β-cells, while no V5 staining was observed in glucagon-secreting α-cells, somatostatin-secreting δ cells, or CD31-positive blood vessels (Fig. 1d). The expression of endogenous TrkB.T1 in β-cells suggests a potential physiological role for TrkB.T1 in these cells.

### TrkB.T1 knockout mice have normal pancreatic development. To study the in vivo role of islet TrkB.T1 receptors, we first investigated whether TrkB.T1 deletion in mouse leads to developmental abnormalities of the pancreas. Immunohistochemical analysis of TrkB.T1 KO and littermate controls (WT) pancreata revealed no differences in total β-cell mass at 3 months of age (Supplementary Fig. 3A, B). Also, no significant differences were found in total insulin content between the two groups (Supplementary Fig. 3E). In addition, immunofluorescence staining of insulin, glucagon, and somatostatin in islets from WT and TrkB. T1 KO mice revealed no quantitative and qualitative difference in islet composition between the two groups (Supplementary Fig. 3C, D). Islet architecture was also not affected by the lack of TrkB.T1 (Supplementary Fig. 3C). The forkhead transcription factor O1 (FoxO1) is involved in the maintenance of β-cell identity and functions, apoptosis, differentiation and proliferation under both physiological and pathological conditions[19]. Thus, we tested FoxO1 expression at various developmental time points in TrkB.T1 and control mice (Supplementary Fig. 3F). Again no obvious difference in FoxO1 expression was observed in all age groups of TrkB.T1 KO and WT mice analyzed[19] (Supplementary Fig. 3F). These data suggest that TrkB.T1 plays no significant role in pancreas development.

### TrkB.T1 KO impairs insulin release and glucose tolerance. We next examined whether TrkB.T1 plays a role in the normal physiology of insulin-producing β-cells. To test this, TrkB.T1-deficient mice were subjected to an oral glucose tolerance test (OGTT). Fasted mice were challenged with a bolus of glucose and blood glucose and insulin levels were measured over a 2-hr time period. As shown in Fig. 2a–d, TrkB.T1 KO mice showed impaired glucose tolerance, associated with blunted insulin secretion. Although no significant changes were observed in the

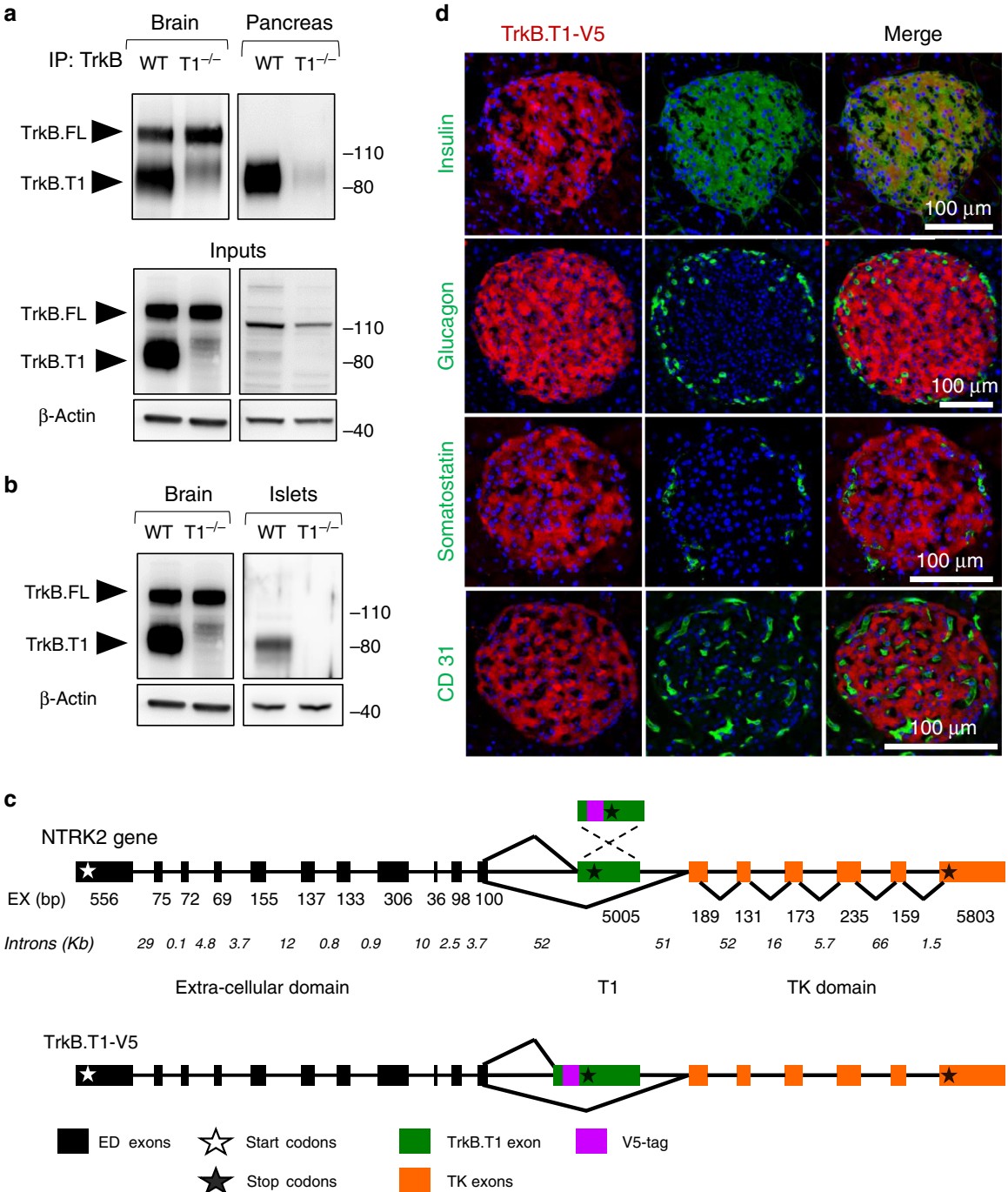

**Fig. 1 TrkB.T1 is expressed in mouse pancreatic β-cells. a**, **b** Western blot analysis of whole mouse pancreas protein lysates immunoprecipitated with a TrkB antibody (**a**) or straight lysates from isolated islets (**b**) blotted with an antibody against the extracellular domain of TrkB to detect all TrkB receptor isoforms. Note that both whole pancreas and isolated islets, express only detectable levels of TrkB.T1. Brain lysates were used as positive control, and TrkB.T1 knockout lysates (T1$^{-/-}$) were used to confirm antibody specificity. On the right is the location of molecular weight markers. **c** Schematic diagram showing the strategy used to tag TrkB.T1 with a V5 epitope to study TrkB.T1 expression in mouse. **d** Immunofluorescent localization of V5-tagged TrkB.T1 in pancreatic islets. Immunostaining for insulin, glucagon, somatostatin and the endothelial marker CD31 shows exclusive TrkB.T1-V5 expression in β-cells. Source data are provided as a Source Data file.

insulin tolerance test (Fig. 2e, f), these data suggest that TrkB.T1 signaling is essential for normal glucose metabolism.

**BDNF enhances glucose-induced insulin secretion from islets**. The metabolic deficits displayed by TrkB.T1 mutant mice suggest a novel role of BDNF in β-cell function. To further dissect this phenotype, we studied insulin secretion in isolated islets in

response to different stimuli. WT islets were cultured overnight in 5.5 mM glucose. The next day, islets were placed in a low (3.3 mM) glucose solution that does not stimulate insulin secretion before testing (Fig. 3a–e). As expected, transfer of islets to high (16.7 mM) glucose induced a significant increase in insulin secretion. However, this increase was dramatically higher when the same islets were washed in 3.3 mM glucose and then re-challenged with 16.7 mM glucose solution containing 1 ng/ml

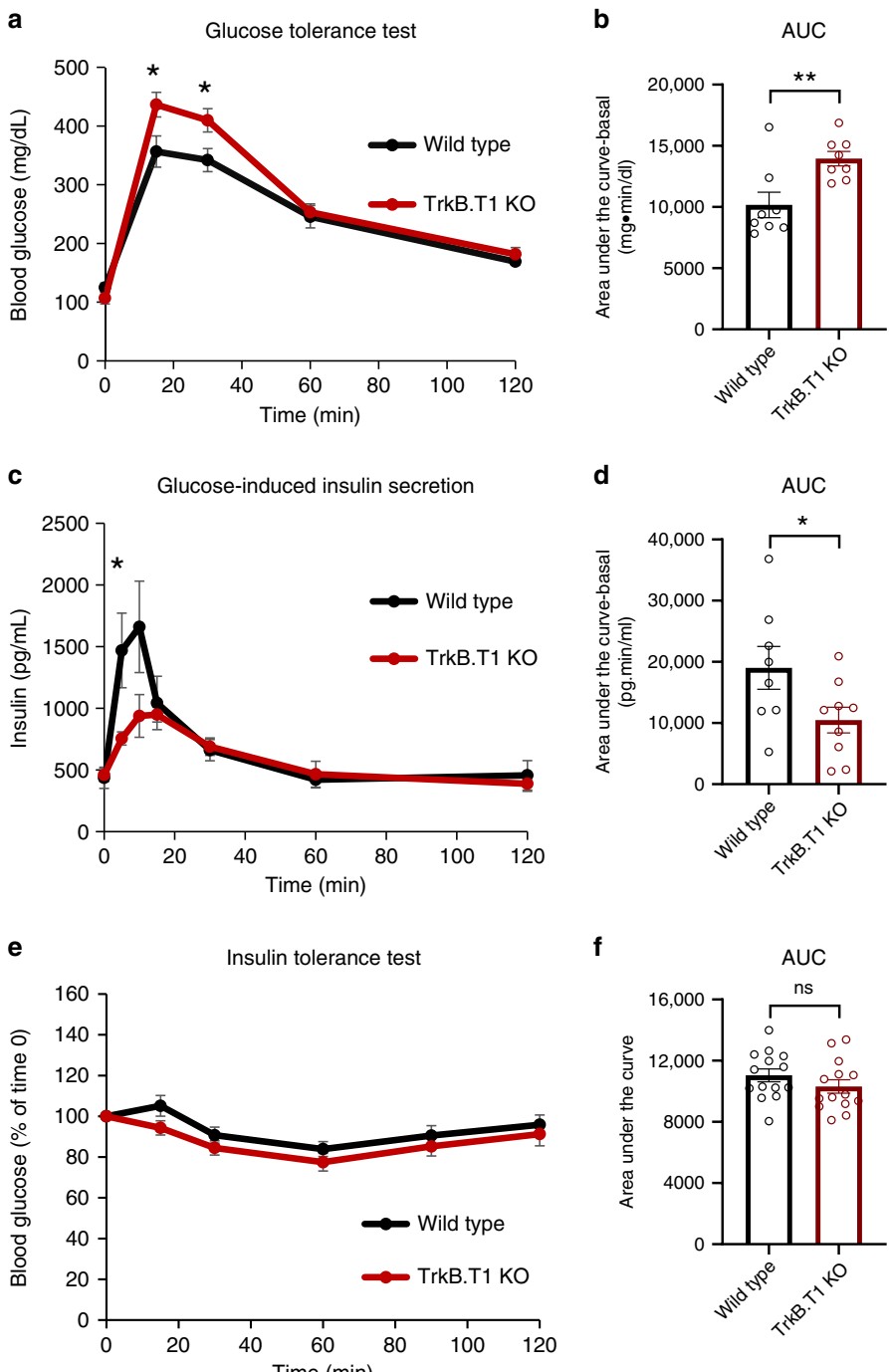

**Fig. 2 Mice lacking TrkB.T1 have impaired glucose metabolism. a–d** Blood glucose (**a**, **b**) and insulin (**c**, **d**) levels in TrkB.T1 KO and control mice subjected to an oral glucose tolerance test (2 g/kg of glucose), WT $n = 8$, TrkB.T1 KO $n = 8$ in **a** and WT $n = 8$, TrkB.T1 KO $n = 9$ in **c**. **b**, **d** Area under the curve (AUC) for the exogenous glucose induced increments of plasma glucose (**b**) and insulin (**d**); *$p < 0.05$, **$p < 0.01$ (Student's $t$-test). **e**, **f** Insulin tolerance test in WT and TrkB.T1 KO mice. Time course (**e**) and AUC (**f**) of blood glucose concentration after a bolus of 0.75 unit/kg insulin was intraperitoneally injected at time 0; WT $n = 14$, TrkB.T1 KO $n = 14$. Data represent mean ± S.E.M. Source data are provided as a Source Data file.

BDNF (Fig. 3a). To investigate if this effect was caused by the consecutive exposure of the islets to two 16.7 mM glucose treatments rather than the presence of BDNF, we performed the same experiment by omitting BDNF in the second treatment. However, the two 16.7 mM glucose treatments caused similar insulin release (Supplementary Fig. 4). To test whether BDNF alone was sufficient to elicit insulin secretion we transferred the islets to a low glucose solution containing BDNF. BDNF alone failed to induce insulin secretion although, first priming the islets with 16.7 mM

glucose led to a small but significant action of BDNF alone on insulin secretion (Fig. 3b, c). Taken together, these data strongly suggest that BDNF is a potent inducer of insulin secretion under hyperglycemic conditions. To test whether BDNF function was transduced by TrkB.T1, we performed the same insulin secretion test with TrkB.T1 deficient islets (Fig. 3d, e). While the mutant islets responded normally to high glucose levels, there was no synergistic action of BDNF with high glucose (Fig. 3d). Moreover, priming the mutant islets with 16.7 mM glucose also did not lead

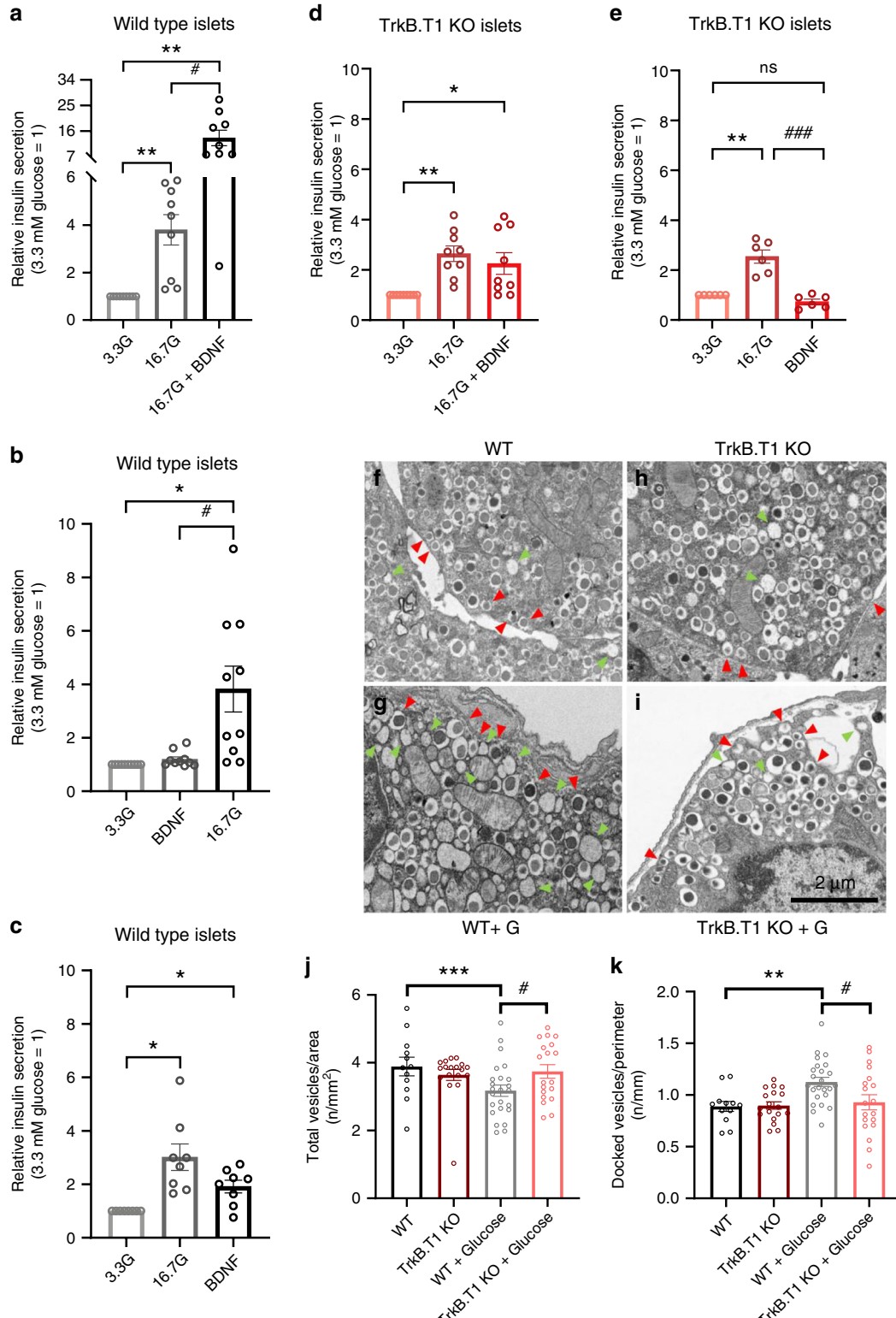

to an increased insulin secretion by BDNF alone (Fig. 3e). These findings indicate that BDNF is a strong facilitator of insulin secretion at high glucose concentrations and that this function is mediated by TrkB.T1.

Next, we investigated if the deletion of TrkB.T1 affected insulin granule secretion. Using electron microscopy (EM) analysis, we quantified the number of insulin granules docked at the plasma membrane following glucose treatment of WT and TrkB.T1 KO mice. Two minutes after glucose i.p. injection, mice were perfused, and pancreata were processed for EM analysis (Fig. 3f–i). While WT mice showed a significant reduction in total β-cell vesicles and an increase in the number of vesicles docked at the plasma membrane, no major changes in total or docked vesicles were observed in TrkB.T1 mutant cells (Fig. 3j, k), again suggesting that TrkB.T1 facilitates insulin release.

**Fig. 3 TrkB.T1 mediates BDNF-induced insulin secretion in islets. a–e** BDNF synergizes with glucose to promote islet insulin secretion; quantification of changes in insulin secretion levels relative to islets treated with 3.3 mM glucose. Conditions are as indicated in panels and include islets incubations for 25 min for each phase with various glucose solutions [3.3 mM (3.3 G) or 16.7 mM (16.7 G) glucose] with or without 1 ng/ml BDNF. Each individual treatment was spaced by a 25-minute wash in a 3.3 G solution. Data represent mean ± S.E.M. of nine independent experiments ($n = 9$) (**a**), $n = 10$ (**b**), $n = 8$ (**c**), $n = 9$ (**d**) and $n = 6$ (**e**). One-way repeated ANOVA *$p < 0.05$, **$p < 0.01$, followed by Tukey's test versus 3.3 G treated phase; #$p < 0.05$, ### $p < 0.001$, Tukey's test versus 16.7G-treated phase (**a, e**) or BDNF-treated phase (**b**). **f–k** TrkB.T1 mutant mice have a reduced number of membrane-docked insulin vesicles in response to glucose. Representative images of transmission electron microscopy of islets from wild-type (WT) (**f, g**) or TrkB.T1 KO (**h, i**) mice injected intraperitoneally with vehicle (**f, h**) or a bolus of glucose (2 g/kg) (**g, i**) and perfused 2 min after injection. Area and perimeter of β-cells were quantified, and total insulin-containing vesicles and insulin-containing vesicles closer than 0.13 μm to the plasma membrane ("docked vesicles") were counted. Red arrows indicate docked vesicles; Green arrows show phantom vesicles after insulin release. **j** Quantification of total insulin-containing vesicles in mice injected or not-injected with glucose ("G"). **k** Quantification of insulin-containing docked vesicles relative to the β-cells perimeter in mice injected or not-injected with glucose. $n = 3$ mice per group. Vesicles counted were from 12 (WT), 18 (TrkB.T1 KO), 24 (WT + G), 19 (TrkB.T1 KO + G) β-cells. **$p < 0.01$, ***$p < 0.001$, followed by Student's *t*-test versus WT group; #$p < 0.05$, followed by Student's *t*-test versus WT + G group. Data represent mean ± S.E.M. Source data are provided as a Source Data file.

**TrkB.T1 signaling increases β-cells cytosolic Ca$^{2+}$ levels**. In astrocytes and cardiomyocytes, activation of TrkB.T1 by BDNF increases intracellular Ca$^{2+}$ levels by inducing Ca$^{2+}$ release from intracellular stores[20,21]. We therefore studied this mechanism in mouse β-cells using a transgenic mouse conditionally expressing the calcium indicator protein GCaMP3[22] in β-cells (Fig. 4a). Pancreata were imaged via confocal microscopy or in a microfluorimetry setup. To visualize whether BDNF had a direct effect on Ca$^{2+}$ levels and to prevent quenching of the signal by high glucose levels we imaged islets at 3.3 mM glucose. BDNF (10 ng/ml) application caused a small but significant increase in β-cell intracellular Ca$^{2+}$ that was completely reversible (Fig. 4b). This effect was totally absent in TrkB.T1 mutant islets, confirming a direct role of TrkB.T1 in mediating BDNF signaling in β-cells (Fig. 4c). In 5.5 mM glucose, most WT islets showed constitutive synchronous Ca$^{2+}$ oscillations (Fig. 4d). Interestingly, adding BDNF increased the amplitude and frequency of Ca$^{2+}$ mobilization (Fig. 4d). In some cases, when islets did not show spontaneous oscillation (4/13 total), BDNF triggered Ca$^{2+}$ oscillations (Fig. 4e). The main frequency of spontaneous oscillations under baseline conditions at 5.5 mM glucose was $1.47 \pm 0.22$ spikes per min and increased to $2.37 \pm 0.31$ upon BDNF application (10 ng/ml for 1 min), an increase of $163.5 \pm 9$ % ($n = 9$; $p = 0.03$).

In human and rodent β cells insulin granule release is triggered by a cytoplasmic increase in Ca$^{2+}$ levels after activation of L-type and P/Q-type Ca$^{2+}$ channels[23]. In addition to plasma membrane Ca$^{2+}$ channels, mouse and human islets express ryanodine and IP$_3$ receptors which mediate intracellular Ca$^{2+}$ mobilization in response to muscarinic receptor activation[24]. In astrocytes, TrkB.T1 has been shown to induce Ca$^{2+}$ release from the intracellular stores through activation of IP3 receptors[21]. Therefore, we tested whether BDNF could induce Ca$^{2+}$ increases in β-cells employing a mouse β-cell line (β-TC-6) as a model system. We found that β-TC-6 cells[25] have a remarkably similar pattern of Trk receptors expression when compared to mouse islets as they express a significant amount of TrkB.T1 and have negligible levels of other Trk receptor isoforms (Fig. 5a, b and Supplementary Fig. 1). Moreover, while BDNF alone does not increase insulin secretion in β-TC-6 cells, it does have a synergistic action with high glucose as in isolated mouse islets (Fig. 5c). BDNF treatment of β-TC-6 cells loaded with the Fluo-4 calcium indicator, induced cytosolic calcium sparks that could not be blocked by the Trk kinase inhibitor K252a (Fig. 5d–f). While the BDNF-induced Ca$^{2+}$ transients persisted in the absence of extracellular calcium, they were eliminated by the sarco/endoplasmic Ca$^{2+}$ ATPase (SERCA) inhibitor thapsigargin, which was indicative of Ca$^{2+}$ release from the intracellular stores (Fig. 5g, h). In addition, pretreatment of β-TC-6 cells with the phospholipase C (PLC) inhibitor U-73122, but not the inactive analog U-73343,

completely eliminated the calcium sparks induced by BDNF (Fig. 5i, j). Taken together, these data suggest that the calcium transients elicited by BDNF-TrkB.T1 signaling in β-TC-6 cells occur through the activation of PLC via a mechanism similar to that operating in astrocytes and cardiomyocytes[20,21].

**Skeletal muscle BDNF is required for glucose tolerance**. Next, we investigated the physiological source of the ligand activating TrkB.T1 in pancreas β-cells. While peripherally expressed BDNF supports organ innervation during development[4], there is no direct evidence that BDNF can be secreted into circulation and act as a hormone or chemokine. In cardiomyocytes, BDNF acts in an autocrine fashion by releasing sarcoplasm Ca$^{2+}$ and increasing cardiac contraction force[20]. Because BDNF synergizes with glucose in the release of insulin (Fig. 3a), we first tested whether BDNF is produced by β-cells to stimulate glucose-induced insulin release in an autocrine fashion. However, deleting BDNF in β-cells with an INS-1 cre transgenic mouse did not impair glucose tolerance (Supplementary Fig. 5). Since it has also been reported that deletion of BDNF in the liver does not impair glucose tolerance[26], we focused on the skeletal muscle and investigated BDNF expression in this organ. To assure a rigorous immunohistochemistry analysis, particularly in muscle, a tissue with high autofluorescence, we generated a mouse model with a knock-in V5 tag in the *BDNF* locus just before the stop codon (Supplementary Fig. 6). Staining of diaphragm and gastrocnemius of BDNF-V5 mice with an anti-V5 antibody showed specific staining in both muscles (Fig. 6a–f). Interestingly, the signal was not uniformly present in all muscle fibers suggesting that within the muscle there is a hierarchy of BDNF expression with some fibers expressing high levels of this neurotrophin and others not expressing it at all (Fig. 6e, f). To investigate if muscle cells can also secrete BDNF we first used the mouse myoblast cell line C2C12, a cell line commonly used as a tool to study muscle biology[27]. RT-PCR analysis shows that differentiated C2C12 cells express *Bdnf* at high levels whereas *Ntf5*, another high affinity ligand for TrkB that has been reported in muscle is expressed at much lower levels[4,28,29] (Supplementary Fig. 7). Importantly, by using a new very sensitive anti-BDNF monoclonal antibody we found that differentiated C2C12 cells secrete BDNF, and this secretion is enhanced by electrical stimulation (Fig. 6g–k). Moreover, BDNF secreted by differentiated C2C12 cells is biologically active as demonstrated by its ability to induce downstream signaling in a reporter cell line that expresses TrkB kinase receptors as sensitive biosensors of BDNF activity[20] (Fig. 6l, m).

To test whether the muscle as an organ can secrete BDNF, we used the mouse diaphragm as a model because it contains significant levels of BDNF and can be easily dissected and

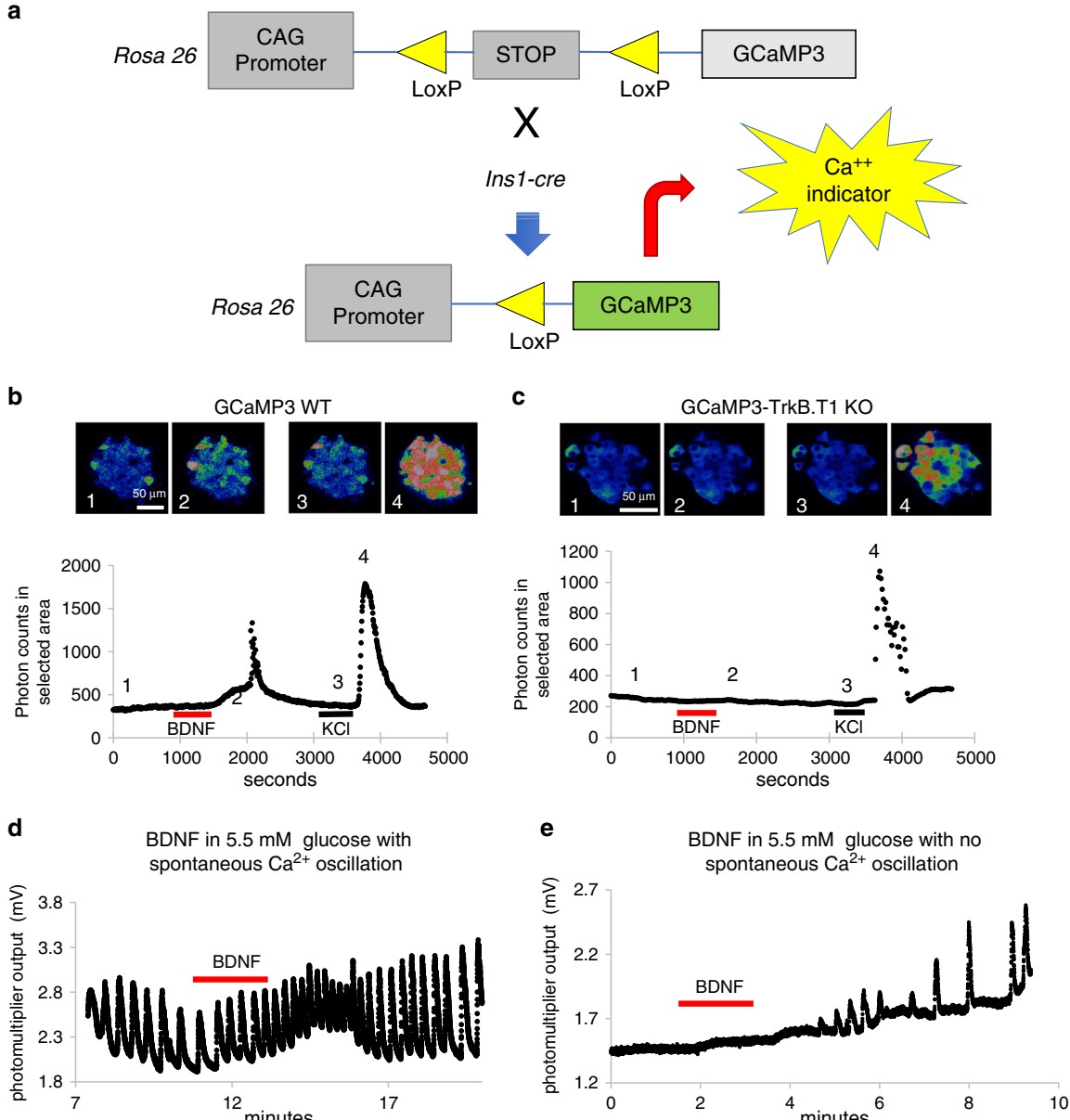

**Fig. 4 TrkB.T1 mediates BDNF-induced Ca$^{2+}$ transients in mouse islets. a** Schematic representation of the strategy used to measure Ca$^{2+}$ levels in β-cells. Intracellular calcium levels were visualized and quantified in islets expressing the genetically encoded calcium indicator GCaMP3 in β-cells using Cre under the insulin promoter (Ins1-cre). **b**, **c** Representative confocal images of WT (**b**) and TrkB.T1 KO (**c**) Ins1-cre-GCaMP3-expressing islets maintained in 3 mM glucose and treated sequentially with 10 ng/ml BDNF (red bar) and 20 mM KCl (black bar) as control. Images were acquired every 2 s, and intensity was plotted (bottom graph). Numbers (1–4) indicate the time at which the images were taken. Note the lack of response to BDNF in the TrkB.T1 KO islets. **d**, **e** Examples of changes in amplitude and frequency of Ca$^{2+}$ oscillation induced by BDNF (10 ng/ml) in 5.5 mM glucose-treated islets. When islets are oscillating, BDNF increases the amplitude and frequency of the oscillations (**d**), whereas BDNF initiates the oscillations when islets are not spontaneously oscillating (**e**).

maintained for ex-vivo preparations due to its reduced thickness (see ref. [30], Fig. 7a). Mouse diaphragms were dissected and electrically stimulated for 1.5 h. The diaphragm supernatant was then applied for 10 min to the cells stably expressing TrkB followed by lysis and analysis for TrkB-activated ERK phosphorylation. Figure 7b shows that supernatants from WT diaphragms induced ERK phosphorylation that could be blocked by pretreatment of the supernatant with a BDNF scavenger TrkB-Fc fusion protein. Interestingly, supernatants from NT5 KO mice diaphragms could also induce ERK phosphorylation comparable to WT ones strongly suggesting that BDNF is the most likely TrkB agonist. Indeed, supernatants from haploinsufficient

BDNF+/− diaphragms[31], or muscle-specific BDNF KO diaphragms (see below) failed to induce ERK phosphorylation further supporting the notion that BDNF is the TrkB ligand secreted from diaphragms (Fig. 7b, c). Interestingly, supernatants from non-electrically stimulated WT diaphragms also induced ERK phosphorylation, although at low levels, suggesting that BDNF can be secreted from skeletal muscle even under resting conditions (Fig. 7c).

To investigate if skeletal muscle-derived BDNF contributes to glucose metabolism, we performed in vivo metabolic studies in a mouse model with the conditional deletion of BDNF in skeletal muscle by the muscle-specific Myl1-cre transgene (Fig. 8a–c).

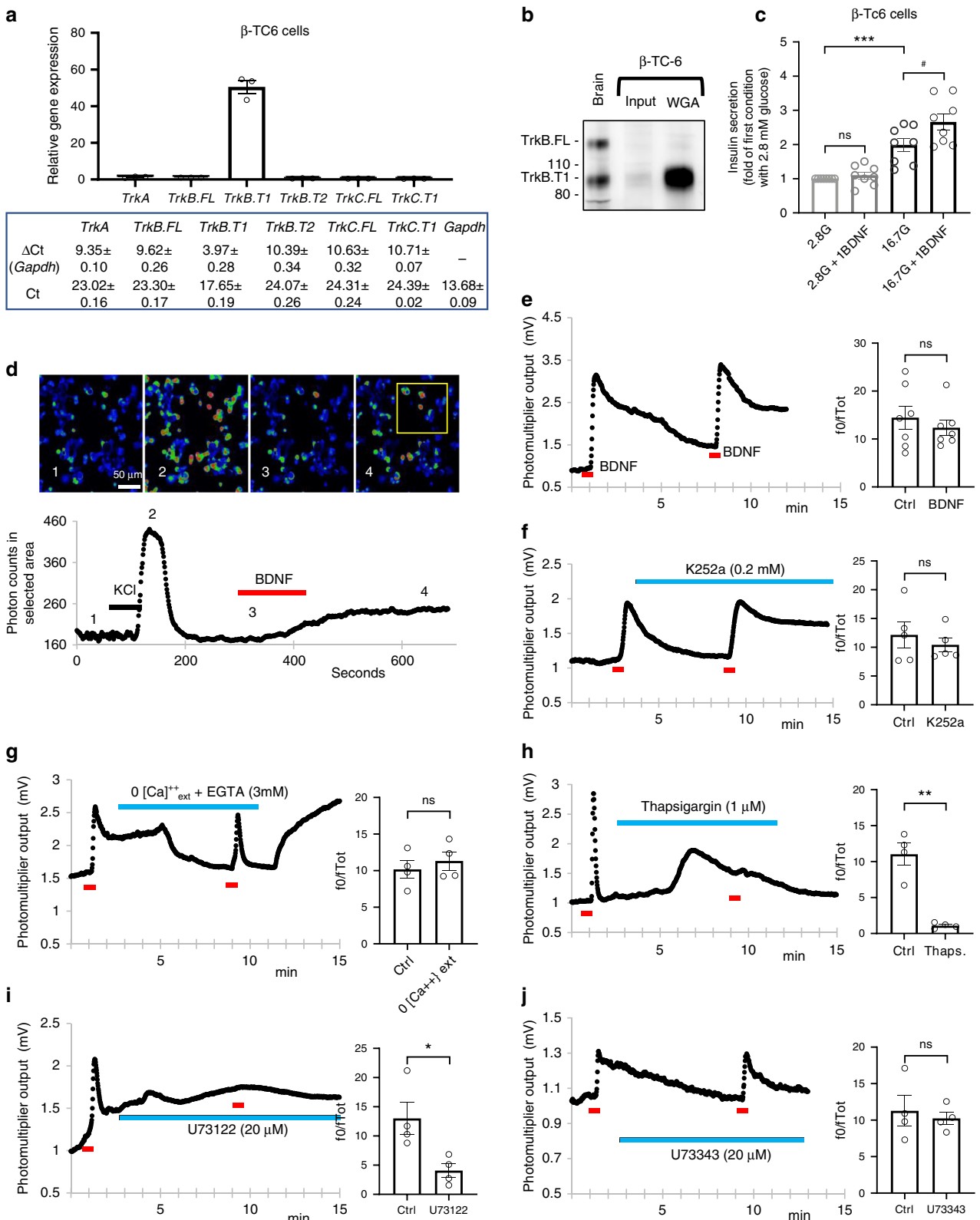

Analysis of Myl1-cre activity was performed using the Rosa26-loxPStoploxP-tdTomato reporter mouse. Strong signal was observed in the skeletal muscle, while spinal cord, pancreas, and hypothalamus showed almost no recombination (Supplementary Fig. 8A). The low level signal present in the lower hypothalamus corresponded to a region with little or no BDNF expression[32]. Indeed, Myl1-cre-BDNF mice showed food intake and body weight comparable to control mice, supporting the specificity of this cre-line and the lack of recombination in hypothalamic regions critical for BDNF regulation of food intake (Supplementary Fig. 8b, c). This is an important finding because, for example, the muscle-specific Myf5-cre line causes some recombination in the brain, leading to mild hyperphagia and weight gain when crossed to a conditional BDNF allele,

**Fig. 5 BDNF induces Ca$^{2+}$ transients in β-TC-6 cells independent of kinase activity. a** β-TC-6 cells express high levels of *TrkB.T1* mRNA. Values are indicated as fold of ΔCt of *TrkB.FL* and *Gapdh* set equal to 1; $n = 3$. **b** Wheat-germ-agglutination (WGA) protein analysis of TrkB expression in β-TC-6 cell lysate showing high levels of TrkB.T1 expression. Brain was used as control for TrkB.FL and TrkB.T1 isoforms expression. **c** BDNF synergizes with glucose in inducing insulin secretion in β-TC-6 cells. Cells were incubated for 30 min with various glucose solutions [2.8 mM (2.8 G) or 16.7 mM (16.7 G) glucose] with or without 1 ng/ml BDNF (1BDNF). One-way ANOVA ***$p < 0.001$, followed by Tukey's test versus 2.8G-treated group; #$p < 0.05$, Tukey's test versus 16.7G-treated group; $n = 8$. **d** Representative confocal live images of β-TC-6 cells loaded with the calcium indicator Fluo-4, treated with 10 ng/ml BDNF and 20 mM KCl (positive control) and imaged live for signal intensity quantification (inset) (**d**) or with a microscope equipped with a microfluorimetry module (**e–j**). BDNF was applied using a fast solution exchanger as indicated (red bar). Twenty micromolar KCl was used as a control of total activation. Numbers (1–4) in **d** indicate the time when the micrographs were taken relative to the Ca$^{2+}$ intensity trace below. **e, f** BDNF causes Ca$^{2+}$ release from β-cells intracellular stores that is not blocked by the kinase inhibitor K252a (**f**, $n = 5$). **g–j** Microfluorimetry analysis of intracellular Ca$^{2+}$ concentrations in β-TC-6 cells treated with EGTA (**g**, $n = 4$), Thapsigargin (**h**, $n = 4$), U73122 (**i**, $n = 4$) or U73343 (**j**, $n = 4$); drug application, blue bar; BDNF (1 ng/ml), red bar. Quantification of the two subsequent BDNF application on the Ca$^{2+}$ induced fluorescence intensity (f0) relative to fluorescence intensity of the permeabilized cells (fTot) is at the right side of each panel. The first bar (Ctrl) indicates the values after the first BDNF application whereas the second bar indicates the values at the second BDNF application (**e**, $n = 7$) or BDNF with the specific drug (**f–j**). *$p < 0.05$, **$p < 0.01$, Student's *t*-test. Data represent mean ± S.E.M. Source data are provided as a Source Data file.

phenotypes which can confound the interpretation of the glucose metabolism results (Supplementary Fig. 8D, E). Mice with skeletal muscle-specific BDNF deletion showed significant glucose intolerance and reduced insulin secretion (Fig. 8a, b). The striking similarity of the metabolic phenotypes between Myl-cre-BDNF KO and TrkB.T1 KO mice strongly suggest that skeletal-muscle derived BDNF is an important activator of pancreatic TrkB.T1 signaling. To further test whether circulating BDNF regulates β-cells activity, we measure plasma BDNF levels in control and muscle-specific BDNF KO mice (Fig. 8d). While, direct measurements of mouse plasma BDNF by ELISA or Western analysis were unsuccessful as previously reported[33], after pooling plasma from multiple animals and immunoprecipitating it with the TrkB-Fc protein we were able to detect BDNF by Western analysis (Supplementary Fig. 6A). To our knowledge this is the first direct evidence that BDNF is present in mouse plasma, though at very low levels. Although, this technique is not quantitative, we found that both at 5 week and 1 year of age, mice with muscle-specific BDNF KO have reduced levels of plasma BDNF (Fig. 8d). To investigate if circulating BDNF can influence insulin secretion in vivo we injected mice with BDNF and tested plasma insulin levels. Testing first for the presence of endogenous BDNF in the pancreas showed that it is present at low levels, and these levels increased following injections of exogenous BDNF (Fig. 8e). However, we found that injections of BDNF did not change circulating insulin levels. Nevertheless, exogenous BDNF, compared to saline injections, induced significantly higher plasma insulin levels after glucose administration (Fig. 8f). Taken together, these data suggest that muscle can secrete BDNF into the circulation and plasma BDNF levels regulate insulin secretion in situations of hyperglycemia as observed in isolated islets (Fig. 3a–e).

**BDNF induces insulin secretion in isolated human islets**. To test whether this BDNF-activated pathway is conserved in humans, we obtained human islets from five non-diabetic donors (Supplementary Table 1). Upon arrival, islets were acclimated overnight in PIM(S) medium containing 5.82 mM glucose. The next day they were transferred to a low glucose solution (1.67 mM) before testing (Fig. 9a–c). Interestingly, treatment of human islets with 1 or 10 ng/ml of BDNF alone caused a significant increase in insulin secretion (Fig. 9a, b). Even more surprisingly, treatment with 8 mM glucose caused an insulin response that was virtually identical to the one obtained with BDNF alone, and no synergy was observed after combining BDNF with high glucose (Supplementary Fig. 9). Further, priming of islets with BDNF did not increase the amount of insulin secretion elicited by glucose alone (Fig. 9a–c and Supplementary

Fig. 9). Taken together, these data strongly suggest that BDNF is a potent inducer of insulin secretion in human islets. To test if this response was mediated by TrkB.T1, isolated human islets, together with mouse and human brain tissues as controls, were subjected to RNA and Western blotting analysis. Figure 9d–f shows that human islets express both truncated TrkB.T1 mRNA and protein[17]. Importantly, when human islets were loaded with Fluo-4 for Ca$^{2+}$ imaging, we found that several cells elicited a rapid increase in cytoplasmic Ca$^{2+}$ levels in response to BDNF, suggesting that these cells contain TrkB receptors (Fig. 9h, i). To test whether TrkB is expressed in human β-cells, we performed immunoelectron microscopy, which allowed us to identify β-cells based on morphology. As shown in Fig. 9g, we found that β-cells have immunogold granules at the membrane, which is indicative of TrkB expression. In summary, these data indicate that truncated TrkB receptors are expressed in human β-cells and transduce the BDNF signal leading to increases in Ca$^{2+}$ levels and subsequent insulin release.

BDNF can be detected at significant levels in human plasma[34]. However, it is unclear whether its presence is due to degranulating platelets during plasma preparation and/or secretion from other organs such as muscle. Therefore, we tested whether human muscle cells can produce and secrete biologically active BDNF. We found that human primary myocytes, after differentiation release BDNF in the culture supernatant and the secreted BDNF is biologically active (Fig. 10a–e). Furthermore, expression analysis showed that human differentiated myoblasts express *NTF4* at negligible levels compared to BDNF suggesting that in both mouse and human BDNF is the predominant TrkB ligand expressed and secreted by muscle cells (Fig. 10f, g).

## Discussion

We have identified a new β-cell signaling pathway regulating insulin secretion in mammals. This pathway includes the anorexigenic neurotrophic factor BDNF as a soluble ligand activating the TrkB.T1 receptor expressed by β-cells. Our data highlight a novel mechanism by which systemic BDNF, in addition to its central metabolic roles, regulates glucose homeostasis through a peripheral site of action. Previous work has shown that systemic administration of BDNF decreases non-fasting blood glucose levels in obese, non-insulin-dependent diabetic mice without significantly reducing food intake[10]. Moreover, 3 weeks of intermittent BDNF administration in an obese diabetic mouse model significantly reduced blood glucose levels[35]. These data suggested a peripheral action of BDNF on blood glucose levels, although the mechanism for this function remained unclear. One obvious question is why, despite the extensive studies on BDNF effects on glucose metabolism, this pathway regulating β-cell

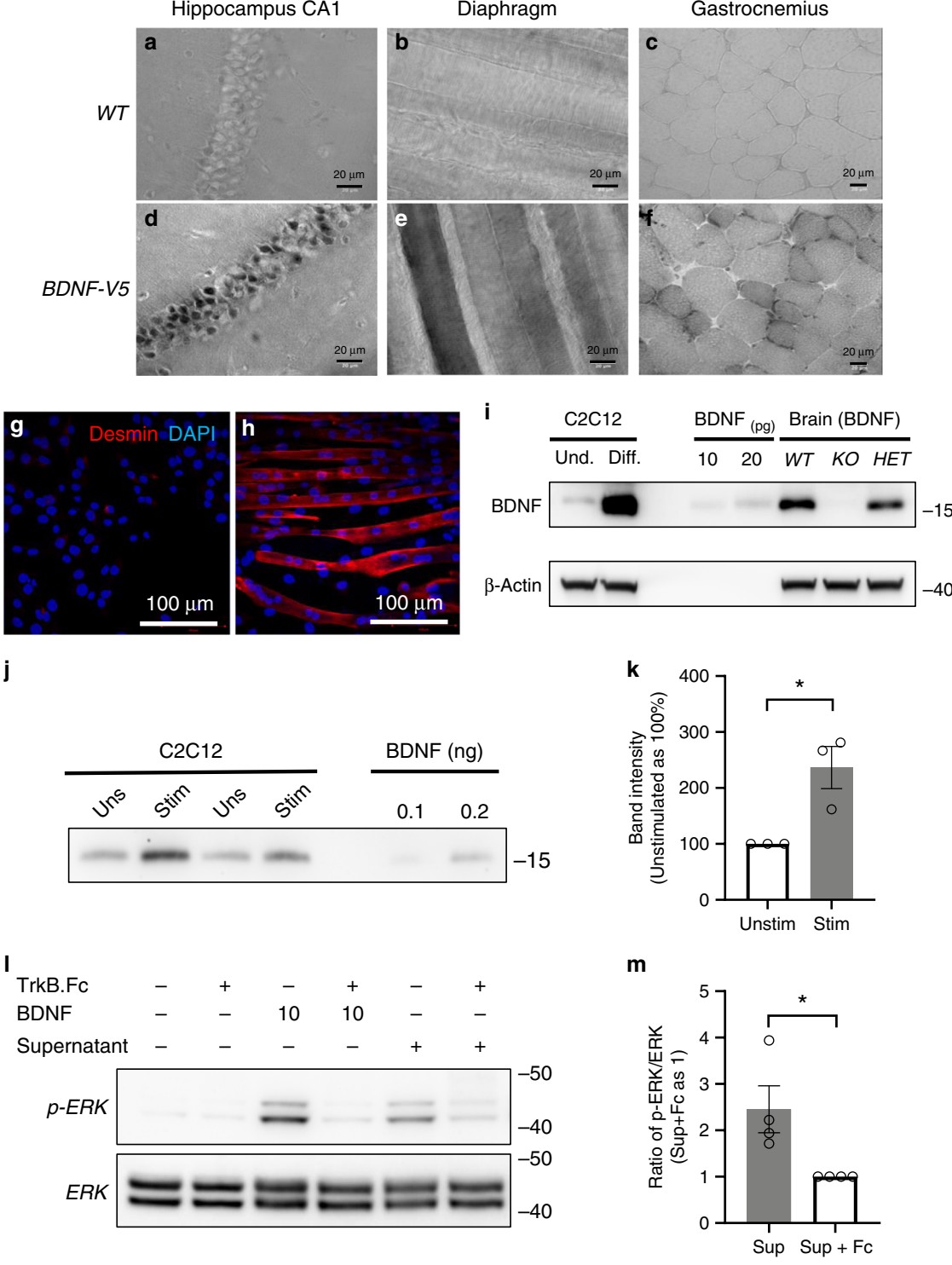

function has remained elusive. There are multiple reasons for this including the relative low levels of TrkB in the pancreas, the lack of sensitive, specific TrkB antibodies useful for immunocytochemistry, and the prevailing notion that truncated TrkB receptors only have an ancillary signaling role relative to TrkB kinase receptors. To circumvent these problems, we generated a mouse model with an endogenously tagged TrkB.T1, thus providing compelling evidence for the presence of TrkB.T1 protein exclusively in β-cells of the pancreas. Furthermore, signaling through this receptor in response to BDNF was further demonstrated by employing a mouse model expressing the calcium

indicator protein GCaMP3 specifically in β-cells to test if TrkB.T1 can increase intracellular $Ca^{2+}$ levels[20,21]. Interestingly, we found that BDNF strongly synergizes with glucose to induce insulin release in mouse islets. This condition had not been tested before, explaining why previous work had failed to detect insulin secretion with BDNF alone[36]. This pathway is conserved between humans and mice; however, a curious difference is that BDNF alone is sufficient to elicit insulin release even at low glucose concentrations in human islets (Fig. 9). This difference could be due to intrinsic properties of mouse and human β-cells including their sensitivity to glucose and expression of different ion

**Fig. 6 Adult mouse skeletal muscle and differentiated C2C12 skeletal muscle cells express BDNF. a–f** Immunohistochemistry analysis of frozen tissue sections from hippocampus (**a, d**), diaphragm (**b, e**) and gastrocnemius (**c, f**) of WT (**a–c**) and BDNF-V5 transgenic (**d–f**) mice immunostained with anti-V5 immunoperoxidase. Note the specific staining in the hippocampus CA1 region (**d**) used as positive control. **g, h** Immunofluorescence staining of undifferentiated (**g**) and differentiated C2C12 (**h**) cells with desmin as control of differentiation. **i** Western blot analysis of undifferentiated and differentiated C2C12 cell lysates immunoreacted with an anti-BDNF antibody (top panel) showing increased BDNF expression after cell differentiation. 10 and 20 pg of recombinant BDNF were used to test antibody sensitivity and brain lysates from WT, heterozygous (HET) and BDNF KO mice were used for antibody specificity. β-actin was used for control of loading. **j** Differentiated C2C12 cells release BDNF upon electrical stimulation. Culture media from stimulated and unstimulated cells for 90 min were immunoprecipitated with a TrkB-Fc protein and immunoreacted for western analysis with an anti-BDNF antibody. Recombinant BDNF was used as control. **k** Quantification of bands as in **j** from three experiments performed in similar conditions; *$p < 0.05$, Student's $t$-test. **l** BDNF released in the culture of differentiated C2C12 cells is biologically active. Media from stimulated C2C12 cells was applied to HEK293 stably expressing TrkB.FL and HEK293 cell lysates were tested for ERK phosphorylation. Treatment of conditioned media with the BDNF scavenger TrkB-Fc protein was used as negative control. **m** Quantification of p-ERK-induced band by conditioned media relative to the same TrkB.Fc-neutralized media, $n = 4$; *$p < 0.05$, Student's $t$-test. Data represent mean ± S.E.M. Source data are provided as a Source Data file.

channels that might influence β-cell sensitivity to intracellular $Ca^{2+}$ levels induced by BDNF/TrkB.T1 signaling[1]. Furthermore, it has been shown that culture conditions can dramatically affect $Ca^{2+}$ oscillations[37]. Thus, β-cell sensitivity to BDNF may be different in mouse islets that were freshly isolated and tested, compared to human islets that had been cultured for several days before delivery. Irrespective, the key finding is that both mouse and human islets express TrkB.T1 and respond to BDNF stimulation with increased $Ca^{2+}$ levels, an important intracellular event triggering the release of insulin.

An important finding in this study relates to the source of BDNF that activates β-cells. Our data suggest that skeletal muscle is a major source of BDNF regulating β-cell activity because skeletal muscle-specific BDNF KO mice phenocopy the metabolic impairments displayed by TrkB.T1 KO mice. Furthermore, we also provided the first unambiguous evidence that BDNF can be produced and secreted by skeletal muscle. BDNF is a contraction-inducible protein in skeletal muscle where it appears to enhance lipid oxidation via AMP-activated protein kinase (AMPK) activation through an autocrine loop[13]. However, until now it remained unclear whether skeletal muscle could produce and secrete biologically active BDNF. In humans, the situation is somewhat more complex as BDNF is stored in large amounts in platelets. Thus, it has been difficult to assess circulating BDNF serum levels because BDNF released from platelets during sampling confounds quantitative analyses[14–16]. Yet, hundreds of studies have investigated serum BDNF levels in humans to test whether changes can be linked to specific pathologies including for example depression[38], cardiovascular disease[39], bipolar disorders[40], obesity[41], and neurodegenerative disorders[42].

Our data provide the most direct evidence to date that BDNF derived from skeletal muscle can regulate blood glucose levels. In addition, we delineated a new mechanism by which BDNF activity can influence mammalian glucose metabolism. Our results support a model in which skeletal muscle activity induces BDNF secretion, enhancing insulin secretion and glucose removal from the blood during hyperglycemia. This mechanism may help explain why physical activity, through the release of BDNF, can normalize hyperglycemia when blood glucose homeostasis is impaired[10,35]. However, during periods of low activity (Fig. 7b, c, unstimulated WT muscle), skeletal muscle may secrete lower amounts of BDNF to sensitize β-cells in preparation of a meal.

In summary, we identified a new BDNF signaling pathway that regulates β-cell function, providing novel insight into the multi-faceted roles of BDNF as a factor regulating mammalian energy and glucose homeostasis. Moreover, these findings together with previous data demonstrating that BDNF regulates cardiac contraction force through TrkB.T1[20], suggest that BDNF may be the factor mediating the positive effect of exercise in improving glucose metabolism and cardiovascular function.

## Methods

**Reagents**. A list of oligonucleotides and antibodies used for the study are listed in Supplementary Tables 2, 3, respectively. K252a (1863), thapsigargin (1138), U73122 (1268/10) and U73343 (4133/10) were from Tocris. Fluo-4-AM (F14201), Powerload (P10020) and Probenecid (P36400) were from Thermo Fisher Scientific. EGTA (E3889) was from Sigma-Aldrich. Brain-derived neurotrophic factor (BDNF) (B-250) was from Alomone labs.

**Mouse models**. Generation of TrkB.T1-V5 and BDNF-V5 mice: sgRNAs for targeting the specific TrkB.T1 exon and the last BDNF exon were designed using the online tool MIT CRISPR Design (crispr.mit.edu). Guide RNAs were generated in vitro using the MEGAshortscript T7 transcription Kit (Thermo Fisher Scientific) and purified by using the MEGAclear Transcription Clean-Up Kit (Thermo Fisher Scientific). A single-stranded DNA donor oligo (150 bp for TrkB.T1-V5 and 182 bp for BDNF-V5) containing the 42 bp V5-tag sequence (14aa: GKPIPNPLLGLDST) preceding the STOP codon was obtained as Ultramer ssDNA oligo (Integrated DNA Technologies). sgRNA (50 ng/μl), ssDNA donor oligo (100 ng/μl) and Cas9 mRNA (100 ng/μl; TriLink Biotechnologies, L-7206) were microinjected into one-cell-stage zygotes obtained from C57BL/6Ncr mice for the generation of TrkB.T1-V5 and BDNF-V5 tagged mice. All mice used for experiments including the TrkB.T1−/− (TrkB.T1 KO)[43]; Neurotrophin 4/5−/− (NT5 KO)[44], Myl1-cre (Jax # 024713), Myf5-cre (Jax # 007893), Ins1-cre (see ref. [45]; Jax # 026801), Gcamp3 (see ref. [22]; Jax # 014538), BDNFloxP/loxP[46] and Rosa26-tdTomato (Jax# 007914) strains were provided with food and water ad libitum and kept in a standard specific pathogen-free environment. All experimental protocols for animal studies were approved by the Committee of Animal Care and Use of the National Cancer Institute in Frederick, Maryland.

**Cell culture**. All media and supplements for cell culture were from Thermo Fisher Scientific and all contained 100 U/ml penicillin, 100 μg/ml streptomycin (15140-122), unless otherwise stated. The β-TC-6 cell line was obtained from ATCC (ATCC® CRL-11506™) and was maintained in low glucose DMEM (11885-084) supplemented with 15% heat inactivated (56 °C for 30 min) fetal bovine serum (FBS) (FB-01, Omega Scientific. HEK293 cells stably expressing the TrkB.FL receptor were maintained in DMEM (11965-092) supplemented with 10% FBS, 2 μg/ml puromycin (A11138-03). Cells were maintained at 37 °C and 5% $CO_2$, and the medium was changed every 3 days.

Sub confluent C2C12 cells (ATCC® CRL-1772™) were cultured in DMEM supplemented with 10% FBS and differentiated for 5 days in DMEM, 2% Horse Serum (Sigma #H1138). Primary adult human skeletal muscle myoblast (HSMM) cells were purchased from Lonza (#CC-2580, Walkersville, MD, USA) and cultured in SkGMTM-2 Skeletal Muscle Cell Growth Medium-2 BulletKitTM (#CC-3245, Lonza). Cells were seeded into 6-well plates at the density of $1.0 \times 10^5$/well, or at $6.0 \times 10^5$ cells/dish in a 10 cm dish. Differentiation into myotubes was performed by culturing the HSMM in DMEM/F12 plus 2% fetal bovine serum for 5 days.

**Western blot and wheat germ agglutinin (WGA) analysis**. Brain, pancreas, β-TC-6 cells and isolated mouse islets were lysed in RIPA buffer supplemented with protease inhibitor (cOmplete™ Mini Protease Inhibitor Cocktail Roche 11836153001) and processed directly for Western blot analysis with the appropriate antibodies or incubated with wheat germ agglutinin (WGA)-agarose beads in RIPA buffer before Western analysis[20]. Blot images were quantified by Image J software (National Institutes of Health). All antibodies and dilutions used in this study are detailed in Supplementary Table 3.

**Immunoprecipitation experiments and human tissues**. Snap frozen ~24 h postmortem, normal human brain sample appearing white matter was obtained from the NIH NeuroBioBank under IRB Exemption number 17-NCI-00070 (NAWM). Human Islets from five different donors (non-diabetic) were provided

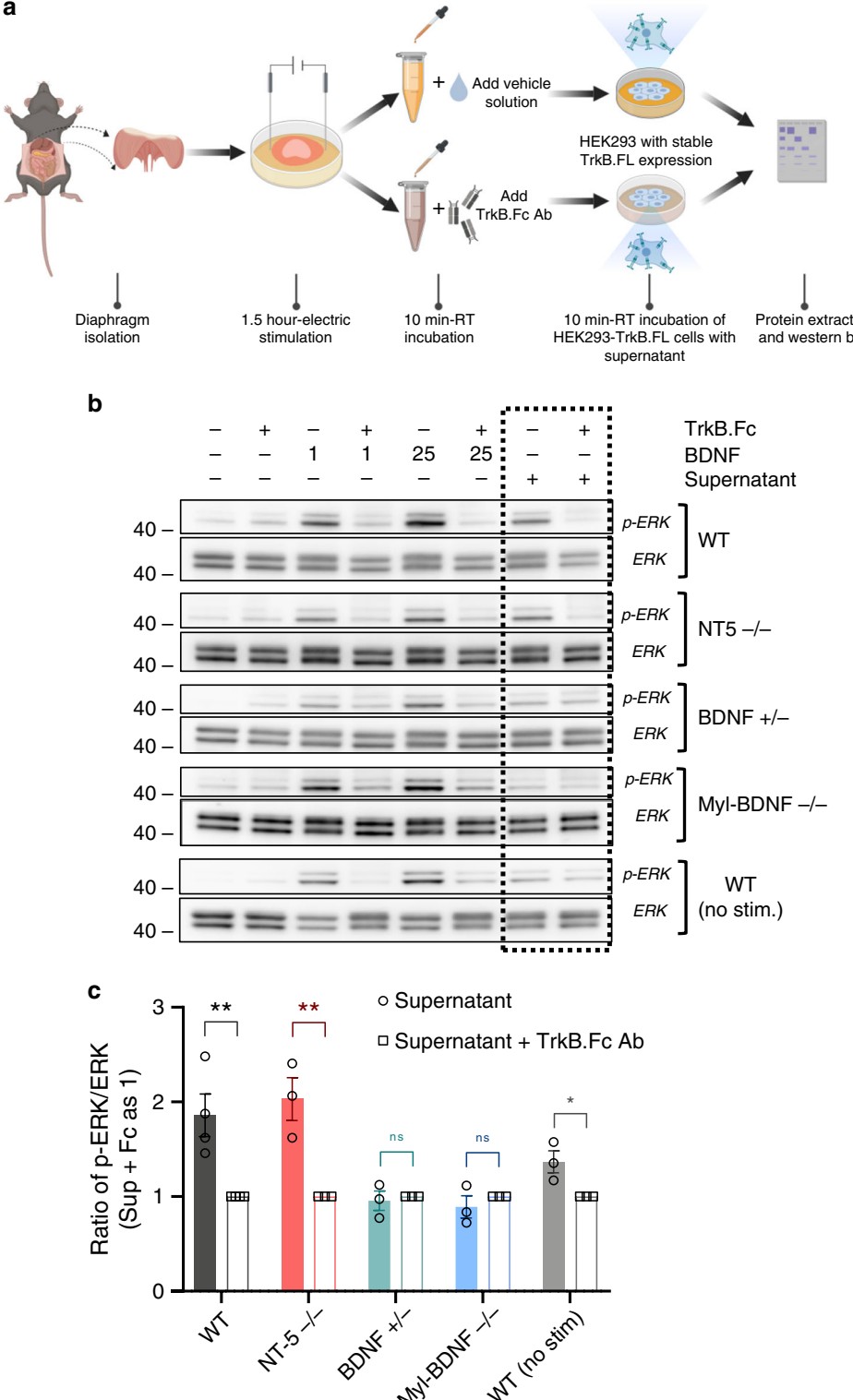

**Fig. 7 Mouse skeletal muscle secretes BDNF. a–c** Skeletal muscle secrete biologically active BDNF. **a** Schematic representation of the procedure used to test diaphragm BDNF secretion and BDNF biological activity. The illustration was created using BioRender (https://biorender.com/). **b** Representative western blot analysis of phospho-ERK (p-ERK) activation relative to total ERK induced by supernatants from isolated mouse diaphragm treated as in **a** and incubated with or without a BDNF-neutralizing TrkB.Fc protein to test for BDNF activity (boxed last two lanes). Cells treated with 1 ng/ml BDNF (lane 3) and 25 ng/ml BDNF (lane 5) were used as positive controls. **c** For each genotype, quantification of supernatant activity (penultimate lane) was performed by setting the ratio of p-ERK/ERK from the supernatant with TrkB.Fc to 1 (last lane), $n = 4$ for WT group, $n = 3$ for NT5−/−, BDNF+/−, Myl-BDNF−/− and WT (no stim.) groups. Student's *t*-test was used for comparison, *$p < 0.05$, **$p < 0.01$. Data represent mean ± S.E.M. Source data are provided as a Source Data file.

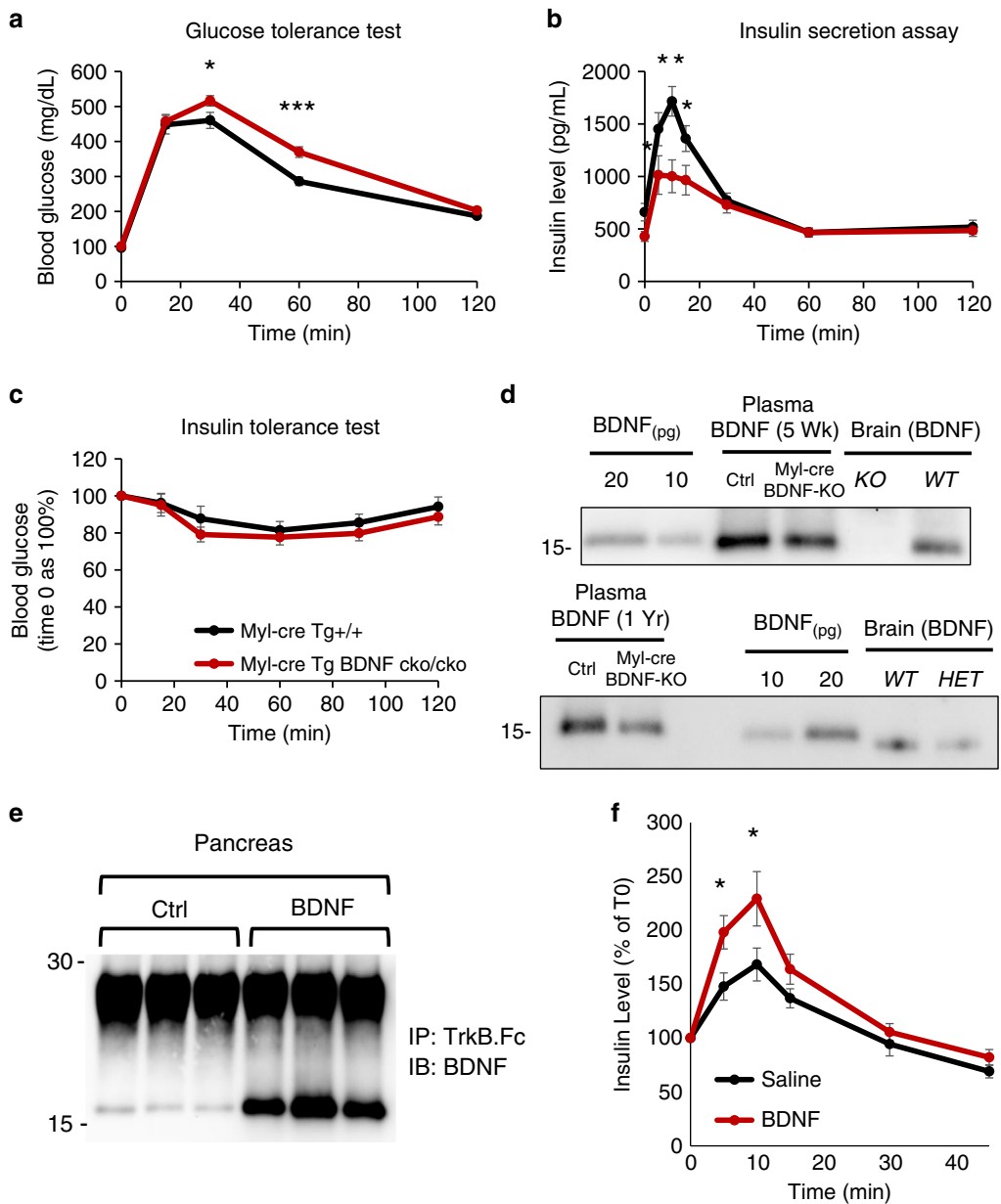

**Fig. 8 Skeletal muscle BDNF is required for normal glucose tolerance and insulin release. a, b** Mice with skeletal muscle BDNF deletion show significant glucose intolerance and reduced insulin secretion. Blood glucose ($n = 16$ mice for both groups) (**a**) and insulin (**b**) levels in Myl-cre Tg BDNF cko/cko ($n = 16$) and control Myl-cre Tg+/+ ($n = 15$) mice subjected to an oral glucose tolerance test (2 g/kg glucose). Myosin light chain driven Cre (Myl-cre) was used to delete BDNF in skeletal muscle of BDNF cko/cko mice. Student's $t$-test, $*p < 0.05$, $**p < 0.01$, $***p < 0.001$. **c** Insulin tolerance test in Myl-cre Tg BDNF cko/cko and control Myl-cre Tg+/+ ($n = 11$ for groups) mice showing no significant difference between groups. **d** Mice with muscle-specific BDNF KO have reduced circulating BDNF levels. Plasma obtained from cardiac punctured blood from 5-week- (5 Wk; top panel) and one year-old (1 Yr; bottom panel) muscle specific BDNF KO (Myl-cre Tg BDNF cko/cko) and control (Myl-cre Tg+/+) mice was pooled from three mice in each group, immunoprecipitated and tested for BDNF by western blot analysis. Recombinant BDNF and brain lysates from WT, BDNF heterozygous (HET) or KO mice were used as control. **e, f** Injected BDNF reaches the pancreas and increases glucose-induced insulin secretion. **e** Pancreas from mice injected intraperitoneally with a bolus of BDNF (100 ng) or saline (Ctrl) were dissected 10 min after injections, washed with saline, lysed, immunoprecipitated with the TrkB-Fc before western blot analysis with a BDNF specific antibody. **f** Plasma insulin levels in mice injected with saline or BDNF as in (**e**) and receiving a bolus of glucose (2 g/kg) by gavage 10 min later. Blood was collected right before the BDNF injection (T0) and 5, 10, 15, 30, and 45 min after the glucose gavage. Student's $t$-test, $*p < 0.05$, $n = 18$ for both groups. Data represent mean ± S.E.M. Source data are provided as a Source Data file.

by the NIDDK-funded Integrated Islet Distribution Program (IIDP) with NIH Grant #2UC4DK098085. According to the new modified policies starting from January 1st, 2019 from NIH Institutional Review Board, no approvals or exemptions are needed for the experiments containing human tissues from deceased and de-identified donors. All donors' information was provided by IIDP program and is listed in Supplementary Table 1. Human islets, fresh mouse tissues, differentiated and undifferentiated C2C12 cells, human HSMM and post-mortem human brain tissues were lysed in RIPA lysis buffer supplemented with proteases inhibitors as

described above. Supernatants were collected after centrifugation (16,000×$g$ × 10 min) and total protein concentrations were determined using the Pierce® BCA Protein Assay Kit (23225, Thermo Fisher Scientific). Dynabeads protein-G (10003D, Thermo Fisher Scientific) were used for immunoprecipitation (IP) and conjugated to the anti-TrkB antibody according to the manufacturer's protocol. IP was performed overnight at 4 °C. Samples were then washed five times with RIPA lysis buffer, resuspended in 2× Laemmli sample buffer (S3401, Sigma-Aldrich) and heated at 95 °C for 5 min before gel analysis.

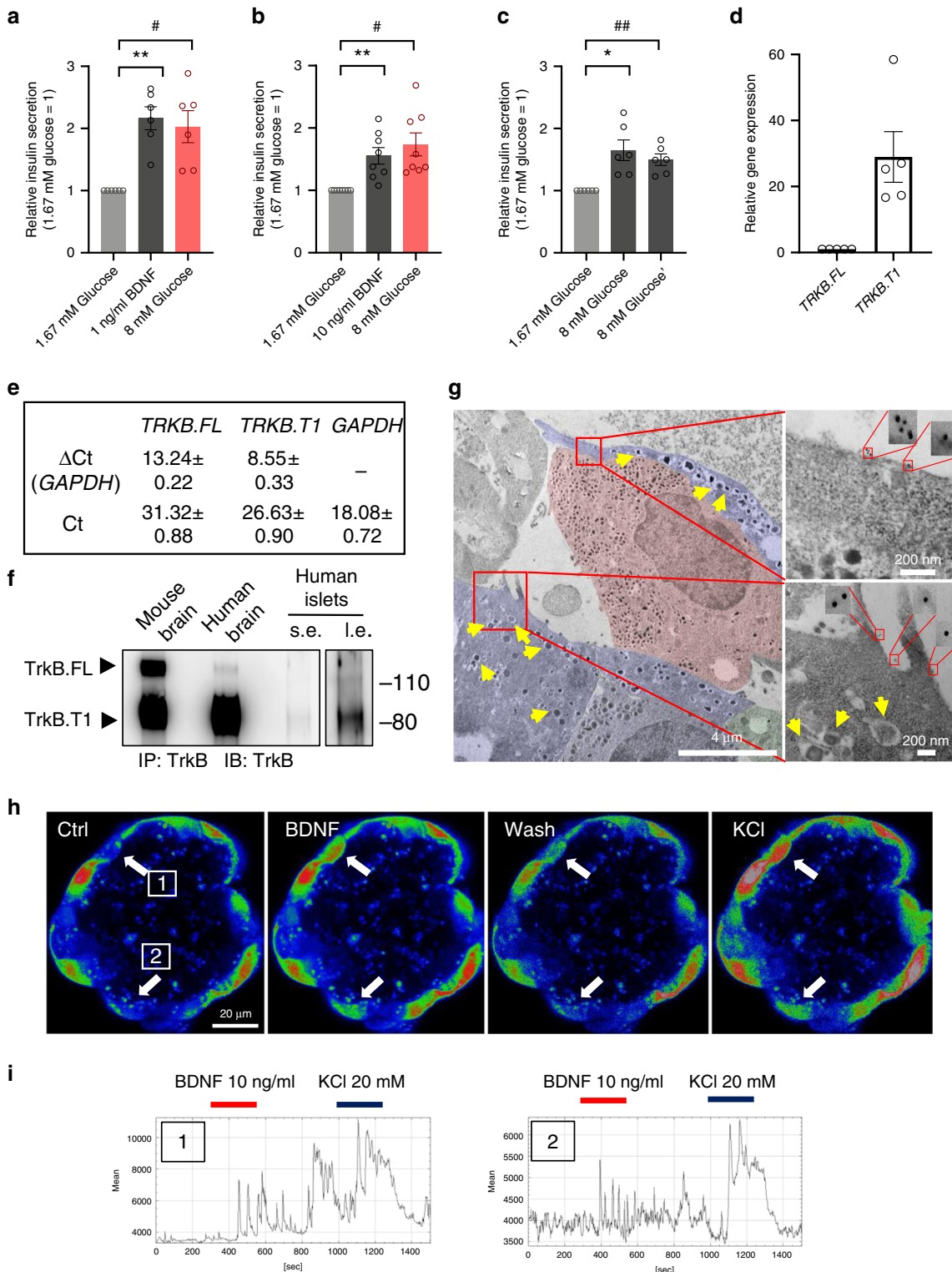

**RNA extraction and quantitative RT-PCR analysis.** Total mouse pancreas RNA was extracted using RNeasy® Plus Universal Mini Kit (QIAGEN) whereas total RNA from isolated mouse islets, human islets, β-TC-6, C2C12 cells, and human HSMM cells, was isolated using the RNeasy Mini kit (QIAGEN) following the manufacturer's recommendations. cDNA was reverse-transcribed using the QuantiTect® Reverse Transcription kit (QIAGEN) and amplified with iTaq Universal SYBR-green Supermix (Bio-Rad Laboratories) in iQ™5 real-time PCR detection system (Bio-Rad Laboratories). PCR products were separated on a 2% (w/v) agarose gel with ethidium bromide and visualized by UV-illumination. The cycling conditions for real-time RT-PCR were: 95 °C for 3 min; 95 °C for 10 s, 60 °C for 30 s for 40 cycles; 95 °C for 1 min followed by the melting curve step at 55 °C (gradient of 1 °C) for 41 cycles. Data were plotted as fold expression relative to the

mRNA level of the *TrkB.FL* isoform. ΔCt values were also calculated using *β-actin* or *Gapdh* as internal control. All PCR primers used in this study are listed in Supplementary Table 2.

**Metabolic studies.** Three-month-old male mice were used for the glucose tolerance test (GTT) and glucose induced-insulin secretion (GIIS) assays. Mice were fasted for 16 h starting at 17:00 pm. The next day at 9:00 am, mice were given 2 g/kg glucose by oral gavage. Blood from tail vein was collected at 0, 5, 10, 15, 30, 60, 120 min after glucose administration using microhematocrit tubes (22-274913, Thermo Fisher Scientific), and glucose concentrations were measured using the Alphatrak®2 glucometer (Zoetis). CritSpin microhematocrit centrifuge (Statspin®,

**Fig. 9 Human islets express TrkB.T1 and respond to BDNF. a–c** BDNF stimulates insulin secretion from human islets. Quantification of changes in insulin secretion levels relative to islets treated with 1.67 mM glucose. Conditions are indicated in the panels and include islet incubations for 25 min in various glucose solutions [1.67 mM glucose (1.67 G) or 8 mM glucose (8 G)] or with 1 or 10 ng/ml BDNF in 1.67 G. Each individual treatment was spaced by a 25-min wash in a 1.67 G solution. Data represent mean ± S.E.M. of multiple independent experiments (**a**, $n = 6$; **b** $n = 8$; **c** $n = 6$) of islets from four different non-diabetic individuals. One-way repeated ANOVA *$p < 0.05$, **$p < 0.01$, followed by Tukey's test versus the 1.67G-treated phase; #$p < 0.05$, ##$p < 0.01$, followed by Tukey's test versus the last 8G-treated phase. **d** qRT-PCR analysis of TrkB isoforms expression in islets from five different non-diabetic donors. Values are indicated as fold of ΔCt of *TRKB.FL* and *GAPDH* set as 1. **e** ΔCt and Ct values used for RT PCR represented in **d**. **f** Western blot analysis of TrkB immunoprecipitation protein lysates from five pooled non-diabetic human donor islets. Mouse brain was used as positive control for TrkB.FL and TrkB.T1. Postmortem human brain lysates were used as a control for antibody specificity. S.e. and l.e.: short and long exposure respectively. **g** Immunoelectron microscopy with a TrkB extracellular domain specific antibody showing 6 nm gold particles exclusively on the surface of human β-cells (insets) identified by the morphology of the insulin vesicles. β-cells are artificially shadowed in blue and insulin vesicle are indicated by yellow arrows. Shadowed in red is an α-cells. **h, i** Human islet cells respond to BDNF with calcium spikes. Islets were loaded with Fluo-4 and imaged during perfusion with BDNF (10 ng/ml) or KCl (20 mM) in 1.67 mM glucose ECS solution. **h** images selected from the time series show cell response (white arrow) to BDNF and KCl (1 and 2). **i** Diagram of Fluo-4 signal intensity over time for the two areas of interest (1 and 2). Source data are provided as a Source Data file.

Thermo Fisher Scientific) were used for plasma collection. Plasma insulin levels were determined by an Enzyme-linked immunosorbent assay (ELISA) kit (80-INSMSU-E01, Alpco) according to the manufacturer's instructions. The area under the curve (AUC) was calculated using the trapezoidal method. The exogenous glucose-induced increments of plasma glucose (0–60 min) and insulin (0–30 min) were expressed as AUC minus basal AUC (basal AUC = fasting glucose value × total time).

For the insulin tolerance test (ITT) mice were starved for 4 h starting at 7:00 a.m. before i.p. injection with a bolus of 0.75 unit/kg Humulin®R U-100 (Eli Lilly). Blood glucose concentration was determined as described above at 0, 15, 30, 60, 90, and 120 min.

BDNF injections to test insulin secretion was done by injecting 3-months-old mice with a bolus of 100 ng BDNF in 100 μl of saline or saline as control i.p. After 10 min mice were subjected to a GTT test as described above. For analysis of BDNF content in pancreas, mice were injected with 100 ng BDNF or saline and after 10 min they were euthanized by CO2 asphyxiation and the pancreas removed, washed in PBS and lysed in RIPA buffer for western analysis as described.

**Calcium imaging in mouse and human islets and in β-TC-6 cells.** Ins1-cre$^{+/−}$; GCaMP3$^{+/−}$ mice expressing the calcium indicator GCaMP3 in beta cells were anesthetized with tribromoethanol (250 mg/kg i.p.), the abdominal cavity was exposed and the bile duct was cannulated to inflate the pancreas with 1.5 % low melting point agarose solution in PBS at 37 °C. After cooling at 4 °C the pancreas was dissected and sectioned with a vibratome at 150 μm. After incubation in ECS [140 mM NaCl, 5 mM KCl, 2 mM NaHCO₃, 1 mM NaH₂PO₄, 1.2 mM MgCl₂, 1.5 mM CaCl₂, 3 mM glucose, and 10 μM HEPES (pH 7.4)] at room temperature (RT), sections with intact islets were anchored to the recording chamber of an Axioskop 2FS with a platinum slice anchor (Warner Instrument), perfused at 1 ml/min at 37 °C with ECS solution, illuminated at 490 nm (LED Thorlabs) and imaged at 40× with a water immersion objective (Zeiss)[47]. Emitted fluorescence was captured by Photodiode (Till Photonics), digitized and stored by Digidata 1322a and clampex software (Axon Instruments). Illumination timing was driven by clampex software, the signal was sampled at 100 Hz, low pass filtered at 50 Hz and stored. In a subset of experiments the recording chamber was placed on the stage of a Zeiss LSM 780 confocal microscope illuminated at 490 nm and imaged through a 20× water immersion objective at 0.2 Hz for off-line analysis.

β-TC-6 cells and human islets, were plated on poly-D-lysine/laminin coated coverslip (Corning), loaded with Fluo-4-AM 1 mM, PowerLoad 100× and Probenecid 1 μM for 30 min at 37 °C (1 h for human islets) before washing with ECS and incubation for 1 h at 37 °C followed by imaging and recording at 40× with a water immersion objective as described above.

BDNF was applied to the preparation with a fast solution exchanger (Warner Instruments) as indicated by the red bar in the corresponding figure. Off-line image analysis was performed with Fiji[48]. Fluorescence induced by BDNF application was quantified integrating the signal of 10 min, for islets, and 2 min for cells using 1-min pre-BDNF application fluorescence as baseline. The calcium ionophore Ionomycin (20 μM) was used at the end of the β-TC-6 cell experiments to calculate the total fluorescence and normalize the response. One-way ANOVA for repeated measure was used for statistical analysis.

**Mouse islets isolation and insulin secretion.** For mouse islets isolation mouse pancreas was inflated through the bile duct with 3 ml ice-cold collagenase solution (0.5 mg/ml collagenase, Sigma-Aldrich, in Medium 199, Sigma-Aldrich, supplemented with 1% FBS)[49]. Dissected pancreas was incubated for 13–15 min in collagenase solution and gently shaken to make the tissue homogeneous. After two washes in ice-cold washing solution (Medium 199, 100 U/ml penicillin, 100 μg/ml streptomycin, 5 mM HEPES; 0.02% -w/v- BSA) and centrifugation at 700×g for 2 min at 4 °C, the supernatant was resuspended in 6 ml Histopaque®-1077 (10771, Sigma-Aldrich). Six milliliter of washing solution were added on top without

mixing before centrifuging at 700×g for 20 min at 4 °C. The supernatant fraction was poured into a Petri dish, 20 ml washing solution was added to further dilute the Histopaque, and the islets were hand-picked with a 20 μl pipette under an inverted microscope before transferring to RPMI 1640 medium (11879-020, Thermo Fisher Scientific) supplemented with 5.5 mM glucose (D9434, Sigma-Aldrich), 10% FBS, 100 U/ml penicillin, 100 μg/ml streptomycin for culturing at 37 °C. After overnight (O/N) culture, about 100 islets from each mouse were picked and placed in a 40 μm nylon cell strainer in a 6-well plate well containing 6 ml of KRBH solution (Hepes-buffered Krebs-Ringer; 125 mM NaCl, 2.56 mM CaCl₂, 5 mM KCl, 1 mM MgCl₂, 25 mM Hepes pH 7.4, oxygenated before use) containing 3.3 mM glucose. Islets were allowed to acclimate for 30 min at 37 °C before transferring to the next well with different solution for testing. The incubation time for each condition was 25 min. Supernatant was then collected for insulin analysis by ELISA kit (80-INSMSU-E01, Alpco) according to the manufacturer's instructions.

For measurement of total insulin content, mouse islets were lysed in an acid ethanol solution (0.4 N HCl, 0.75% ethanol) left rocking O/N at 4 °C. Insulin content was determined by ELISA kit as above and total DNA was determined by Quant-iT™ PicoGreen® dsDNA Assay Kit (P7589, Thermo Fisher Scientific). Total insulin content was expressed as total amount of insulin normalized to total DNA concentration.

**Human islets insulin secretion.** Human islets from the IIDP were cultured O/N for recovery in PIM(S) medium (Prodo Lab) before testing. Insulin secretion experiments were performed with about 80 islets in Krebs solution (114 mM NaCl, 5 mM KCl, 24 mM NaHCO₃, 1 mM MgCl₂, 2.2 mM CaCl₂, 1 mM KH₂PO₄, 10 mM HEPES, pH 7.4) oxygenated for 15–20 min at RT and equilibrated for 15 min at 37 °C[49]. Testing was performed as for mouse islets.

**Insulin secretion measurement in β-TC-6 cells.** For insulin secretion experiment, β-TC-6 cells were seeded at $0.25 \times 10^6$ per well in a six-well plate. The second day, cells were first washed twice with the KRBH solution (140 mM NaCl, 5 mM KCl, 2 mM NaHCO₃, 1 mM NaH₂PO₄, 1.2 mM MgCl₂, 1.5 mM CaCl₂, 2.8 mM glucose, 10 mM HEPES, and pH 7.4) and pre-incubated in the same solution for 30 min. The cells were then incubated for 30 min in 2.8 mM glucose-KRBH, 2.8 mM glucose-KRBH + 1 ng/ml BDNF, 16.7 mM glucose-KRBH, or 16.7 mM glucose-KRBH + 1 ng/ml BDNF. Supernatants were collected for insulin analysis by ELISA kit (80-INSMSU-E01, Alpco) according to the manufacturer's recommendations.

**Physiological studies with mouse diaphragms.** Mouse diaphragms were quickly removed, cleaned with D-PBS, and fixed in a sylgard-coated dish with 4 ml of ECS solution, placed inside a closed chamber half filled with ECS and bubbled with 95% O₂ and 5% CO₂. Stimulations of 1 s long train at 100 Hz (stimulus duty cycle 1/10 or 100 ms) and every 10 s for a total of 90 min were applied to the isolated diaphragm through stainless steel electrodes placed on the sides of the diaphragm muscle. The stimulation current was set to 10 mA. Supernatants were collected and incubated with or without TrkB-Fc antibody (688-TK, R&D Systems) at room temperature for 10 min. Supernatants with or without TrkB-Fc antibody were then incubated with HEK-293 cells stably expressing the TrkB.FL receptor for 10 min at 37 °C. Subsequently, cells were lysed and analyzed by Western blot analysis.

**BDNF secretion from C2C12 and human myoblast cells.** Media from differentiated and undifferentiated murine C2C12 cells or differentiated HSMM was harvested after for 4 or 6 days in culture, respectively, centrifuged and incubated with or without the TrkB.Fc antibody for 5 min at room temperature. The media was then immunoprecipitated and analyzed by western blot as described above. The biological activity of the media was tested by exposing HEK-293 cells stably

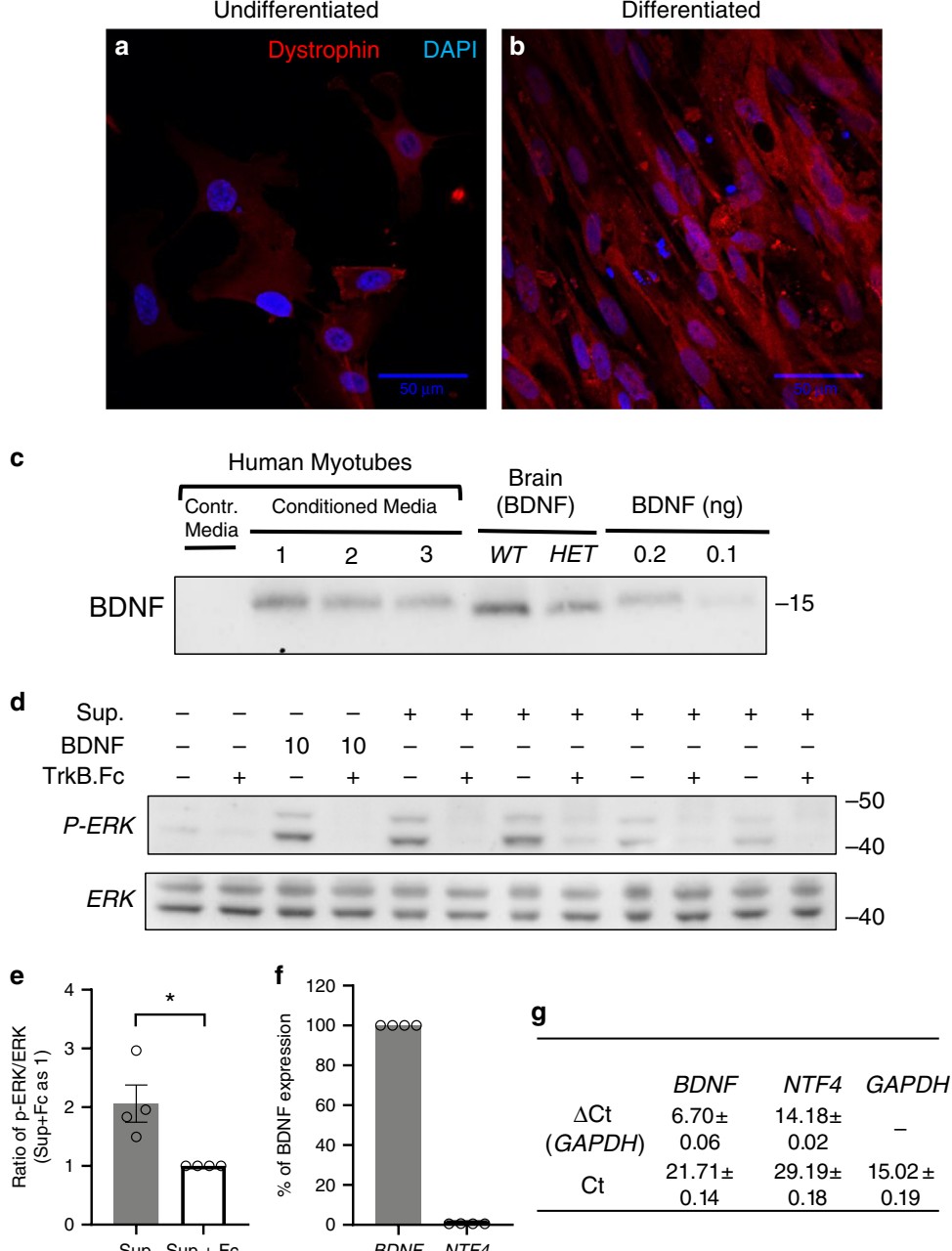

**Fig. 10 Human primary skeletal muscle myoblasts secrete biologically active BDNF. a–g** Adult, human primary skeletal muscle myoblasts (**a**) differentiated in vitro for 5 days express dystrophin (**b**). **c–e** Differentiated human myotubes secrete biologically active BDNF. **c** Western blot analysis of conditioned media from three independent cultures (1–3) immunoprecipitated with the TrkB-Fc protein and immunoblotted with an anti-BDNF antibody. Equal amounts of brain lysates from WT and Heterozygous (HET) mice were used as control. Recombinant BDNF was used to test sensitivity. **d** Conditioned media from differentiated human myotubes was tested for BDNF biological activity as in Fig. 6l and quantified from four independent cultures in **e** Student's $t$-test, $*p < 0.05$. **f, g** qRT-PCR analysis of BDNF and NTF4 expression in human myotubes showing negligible levels of NTF4 mRNA expression compared to BDNF ($n = 4$). Data represent mean ± S.E.M. Source data are provided as a Source Data file.

expressing TrkB.FL receptor to the media for 10 min. HEK-293 cells were then lysed and phosphorylated and total ERK was analyzed by Western Blot.

**Immunofluorescence and immunohistochemistry staining**. For immuno-fluorescent staining experiments, mice were transcardially perfused with 4% par-aformaldehyde (PFA) (w/v). Brain, pancreas, spinal cord, and skeletal muscle were carefully removed and immersed in 4% PFA for an additional 24-h post-fixation at 4 °C, followed by 48-h cryoprotection in 30% sucrose (w/v) at 4 °C. Frozen samples were sectioned at 15 μm. Cells plated on glass coverslip were washed twice in PBS, fixed with 4% PFA for 1 h at room temp and washed again twice in PBS. Sections or coverslips were incubated in blocking solution (D-PBS, 10% normal donkey serum (NDS), 0.1% Triton X-100) for 1 h at RT before O/N incubation at 4 °C with

the primary antibodies diluted with blocking solution. After incubation for 1 h at RT with the appropriate secondary antibody, washing and counterstaining with 0.25 μg/ml DAPI, sections were mounted with fluorescence mounting medium (Dako) and imaged in a LSM 780 confocal microscope (Zeiss).

For V5 skeletal muscle immunohistochemistry BDNF-V5 and WT control mice were killed by CO2 asphyxiation, muscles were excised and snap frozen in isopentane cooled with Liquid $N_2$. Fifteen micrometer cryostat sections from mutant and control muscles were cut, placed on the same slide, air dried and fixed with 2% paraformaldehyde in PBS for 5 min. Endogenous peroxidases were blocked with 0.3% $H_2O_2$ in PBS for 5 min and in 2% NDS in PBS for 10 min. Primary rabbit antibody against the V5 epitope was applied at 1:100 in PBS with 2% donkey serum for 25 min followed by the secondary biotinylated antibody against rabbit IgG before 3, 5 min wash in PBS + 2% NDS at 1:500 in the same

media for 25 min. The ABC peroxidase kit (Vector laboratories #SK4100) was used to develop the sections. Slides were imaged at 40× and captured with an ORCA-ER camera (Hamamatsu) using Zeiss Zen-Blue software.

**Pancreas islet content and quantification.** Number, size, and area occupancy of islets was obtained from fixed pancreas flat-embedded in paraffin and sectioned at 10 µm collecting 2 adjacent section every 8 throughout the whole pancreas. One every 8 sections was incubated with 0.3 % $H_2O_2$ for 30 min at RT to block endogenous peroxidases then with D-PBS supplemented with 10% NDS, 0.1% Triton X-100 for 1 h at RT, following by O/N incubation with the anti-insulin primary antibody at 4 °C followed by a biotinylated horse anti-mouse IgG (BA-2000, Vector Laboratories) secondary antibody for 1 h at RT. After incubation with the ABC (Avidin Biotin Complex) solution for 1 h at RT, Insulin positive cells were detected with the Sigma Fast$^{TM}$ 3,3'-diaminobenzidine (DAB) solution (Sigma-Aldrich). The second section was processed for Masson's trichrome staining. Insulin and trichrome stained sections were digitized with an aperio slide scanner at 20× magnification and analyzed using ImageScope software (Leica and ImageJ). Automated quantification of β-cell mass was performed by the procedure of Golson et al.[50].

**Electron microscopy analysis.** For transmission electron microscope (TEM) analysis of mouse islets insulin secretion, mice were anesthetized with tri-bromoethanol (250 mg/kg) and injected i.p. with a 2 g/kg bolus of glucose in saline. After 2 min mice were perfused for 30 sec with PBS followed by 5 min with fixative solution (2.5% glutaraldehyde, 0.5% tannic acid, 30 mM sucrose in 0.1 M caco-dylate buffer). The pancreas was then removed and post-fixed for 2 h in the same fixative followed by 1 h in 1% osmium tetroxide buffer. Small trimmed tissue blocks of about 1 mm$^3$ were washed three times in buffer and stained with 2% uranil acetate in 50% ethyl alcohol for 1 h, dehydrated and embedded in Epon-Araldite resin per standard protocol. Sixty nanometer ultrathin sections were stained with lead citrate and imaged with TEM, (Technai T12 FEI at 1900× magnification). Analysis of cell profile dimension and insulin vesicles content was performed using ImageJ software.

For immuno-EM of human islets, approximately 30 islets per condition were processed. After washing in PBS, islets were fixed in 2% PFA, 0.5% glutaraldehyde PBS solution for 30 min, washed in PBS and blocked with 2.5% NDS in PBS for 10 min. The primary TrkB goat antibody was applied at 1/200 in PBS, 2.5% NDS for 2 h at RT. After 3, 10 min wash in PBS, islets were treated with donkey anti goat 6 nm gold conjugated secondary antibody at 1/100 in PBS O/N at 4 °C in a rotating drum. After 3, 10 min washes in PBS islets were embedded in 2% low melting agarose at 37 °C, centrifuged and placed in ice for 10 min. Pellets from agarose gel were trimmed and placed in 2% uranyl acetate in 50% ethanol water mixture for 1 h, dehydrated and embedded in Epon/araldite resin. Semithin (0.5 microns) and ultrathin (60 nm) sections were cut and imaged in a TEM at 100 KV.

**Statistical analysis.** Data represent mean ± S.E.M. For statistical analysis of two groups Student's *t*-test was used for comparison (Figs. 2, 3, 5–10 and Supplementary Figs. 3, 5, 8). One-way ANOVA or one-way repeated ANOVA followed by Tukey's test was used for analysis of multiple groups (Figs. 3, 5, 9 and Supplementary Figs. 4, 9). Data were plotted by GraphPad prism 8.0.

## Data availability

The source data underlying Figs. 1a, b, 2a–f, 3a–k, 5a–c, 5e–j, 6i–m, 7b, c, 8a–f, 9a–f, 10c–g and Supplementary Figs. 1A-C, 2A, 3B, 3D, 3E, 4, 5A-D, 6A, 6C, 7A-B, 8B-E, 9, are provided as Source Data File including the original EM pictures, uncropped blots and all reported averages in graphs.

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

## Acknowledgements

We thank Kim Peifley and Stephen Lockett of the Optical Microscopy and Analysis Laboratory for their support with the confocal analysis, Lu Zhu, Rejji Kuruvilla and Jaroslawna Meister for technical suggestions, James Dunleavey for help with the human brain samples, Yang Feng for reagents, Alejandro Caicedo for technical help on human islets imaging, the NIDDK-funded Integrated Islet Distribution Program (IIDP) for providing the human islets and the Tessarollo's lab for comments, suggestions and critical reading of the manuscript. This work was supported by the Intramural Research Program of the NIH.

## Author contributions

G.F., Z.H., F.T.A., J.B., C.B., D.S., and S.Y. performed experiments; G.F., Z.H., L.F.B., S.Y., B.S.C., J.W., O.G., and L.T. provided reagents, intellectual input and scientific expertise; G.F., Z.H., and L.T. wrote the paper.

## Competing interests

The authors declare no competing interests.
