## [Peer Review File · Nature Communications]

Reviewers' Comments:

Reviewer #1:

Remarks to the Author:

This is an interesting and important study, which defines BDNF as a myokine and suggests that muscle-derived BDNF induces insulin secretion. With regards to pancreas, human islets were included. The manuscript is very well written, and the conclusions are fair. The weakest part is the muscle work and the lack of exercise interventions.

I have some suggestions and/or questions that need to be addressed.

1. It would be warranted to include studies on resting and contracting quadriceps muscle. And it would strengthen the study if human muscle cells could be included.
2. A skeletal muscle immuno staining BDNF panel including Diaphragm and gastrocnemius muscle (or other muscle type used during exercise) is needed to show that 1) BDNF is expressed in the muscle and not in other cell types in diaphragm 2) BDNF is expressed in a muscle type that is directly affected by exercise.
3. It would be warranted to show that BDNF infusion (in vivo) stimulates insulin secretion and whether BDNF exerts its effects in a dose-dependent fashion?
4. I miss measurement of insulin sensitivity in the various mice models.
5. It appears that the muscle BDNF knockout mouse has a lower fasting glucose? Why?
6. Finally, it would be interesting to do exercise studies (wheel running or treadmill running) in the various mouse models and to study glucose and/or insulin levels to actually prove that BDNF is involved in a muscle-pancreatic endocrine loop.

Minor comments

The scales are different in figure 1D (Merge), most likely due to different islet sizes and that the pictures have been adjusted. Please check the scales, write in the legend what the scale corresponds to if you don't add it to all the pictures. In picture Insulin/TrkB.T1-V5 merge, only part of the unit is visible. Please adjust.

The specificity would have been even more convincing if you had added BDNF alone without glucose in Figure 2H (KO of the TrkB.T1 receptor).

It would be good to have a y-axis for the calcium oscillations, as the authors have included in figure 5. It is difficult to compare the magnitude of the oscillations with other studies. It is not clear how you have performed the statistics in this analysis.

The authors have included a useful qPCR panel over the expression of the different splice variants of TrkA, TrkB and TrkC in beta-tc-6 cells. However, when using UCSC in silico qPCR blast and NIH primer blast the used primers for TrkC.FL or TrkC.T1 do not have any template. That could explain why you apparently don't get any signal. Please correct.

In relation to that, you have included TrkB.T2 mouse primers in the primer table but do not include any TrkB.T2 measurements (and no specific product can be detected here either). Please only include what you have measured. NB: check supplementary
Also, please add the CT value range for TrkB.T1 for mouse and humans. It would be good to know the approximate expression levels.

Reviewer #2:

Remarks to the Author:

Communication between insulin target tissues and beta-cells is at the heart of beta-cell compensation for insulin resistance. This paper proposes the novel idea that BDNF is secreted from muscle and then binds to the TrkB.T1 receptor in β -cells, leading to the stimulation of insulin secretion. Notably, knockout of BDNF expression in skeletal muscle, but not beta-cell produced glucose intolerance, pointing to skeletal muscle as the relevant source of the BDNF that regulates beta-cells. This is an important and novel finding.

1. The data presented in Figures 3H-3M are not accompanied by a quantitative data.
2. The insulin responsiveness of the β -TC6 cell line is not shown; i.e. data comparable to that presented in Figs 2E-2H.
3. The second cell in Figure 5H appears not to be a beta-cell; in contrast to the first cell, Ca oscillations in this cell seems to fall off after treatment with BDNF.

Typos

1. Line 231 "following" should be "followed"
2. Line 272 "where" should be "were"
3. Line 291: "in obese diabetic mice model" should be "in an obese diabetic mouse model"

Reviewer #3:

Remarks to the Author:

The manuscript by Fulgenzi et al. describes a novel mechanism by which the skeletal muscle regulates insulin secretion. The authors found that the truncated TrkB receptor (TrkB.T1) was expressed in beta cells of the pancreas in both mice and humans. They showed that activation of TrkB.T1 by BDNF enhanced glucose-induced insulin secretion in murine beta cells and directly stimulated insulin secretion in human beta cells. Removal of TrkB.T1 abolished the BDNF effect on insulin secretion. To support these in vitro findings, the authors further showed that mice lacking TrkB.T1 displayed impairments in glucose tolerance and glucose-induced insulin secretion. Additionally, the authors found that muscle-specific Bdnf knockout mice also showed impairments in glucose tolerance and glucose-induced insulin secretion. Therefore, the authors conclude that muscle-produced BDNF binds to TrkB.T1 in beta cells to promote insulin secretion. In general, the manuscript is well written, and the presented data are solid. The study not only uncovers a regulation mechanism of insulin secretion and a novel function for the truncated TrkB receptor, but also provide insights into the metabolic benefit of exercise. Thus, the manuscript is interesting to a broad readership. The following concerns should be addressed before the manuscript is published.

1. Additional evidence would be helpful to link muscle BDNF to TrkB.T1 in beta cells. BDNF produced in the muscle should reach the pancreas through the circulation. If the authors' model is correct, levels of BDNF in the circulation and the pancreas should be reduced in muscle-specific Bdnf knockout mice. Alternatively, infusion of BDNF into the circulation should rescue the glucose-related phenotypes in the knockout mice.
2. The authors point out that TrkB antibodies are not specific enough for immunohistochemistry. Instead, they used the antibodies for immunoelectron microscopy. The logic does not sound correct, because immune EM is one type of immunohistochemistry. The authors can perform in situ hybridization using ViewRNA or RNAscope in situ kits to show specific expression of TrkB.T1 in human beta cells.
3. Line 246: The statement "while spinal cord, pancreas and hypothalamus showed almost no

recombination (Suppl. Fig. 6A)" is not accurate. Supplementary Fig. 6A shows significant recombination in the arcuate nucleus of the hypothalamus. The arcuate nucleus is an important brain site for the regulation of glucose homeostasis. To lessen the concern, the authors could cite published studies that show little or no expression of BDNF in this hypothalamic region.

4. The magnification in Figures 4I-L is too low to see insulin vesicles.

Thank you very much for the constructive comments that were provided by the whole reviewing team. We have now experimentally addressed all issues to strengthen and improve our manuscript. A point-by point rebuttal to all issues raised by the reviewers is below in red. Changes are also reported in red in the manuscript.

Reviewer #1 (Remarks to the Author):

This is an interesting and important study, which defines BDNF as a myokine and suggests that muscle-derived BDNF induces insulin secretion. With regards to pancreas, human islets were included. The manuscript is very well written, and the conclusions are fair. The weakest part is the muscle work and the lack of exercise interventions.

We thank the reviewer for the positive feedback and the comments about the muscle work of the study. In this regards we have done a number of experiments aimed at strengthening the data on the muscle.

I have some suggestions and/or questions that need to be addressed.

1. It would be warranted to include studies on resting and contracting quadriceps muscle. And it would strengthen the study if human muscle cells could be included.

This is a valid point. The reason we used the diaphragm for the ex vivo paradigm was a practical one. This striatal muscle is thin and has a high surface/volume ratio that allows to collect the low level of BDNF produced by almost all muscle cells. Unfortunately, this is not the case for quadriceps muscles that have a very limited surface preventing most fibers from being exposed to the media. Furthermore, the stimulation in diaphragm was done over a period of 1.5 hours, which would result in necrosis of the larger quadriceps due to the inability to receive the necessary nutrients ex vivo. However, to strengthen the point of BDNF secretion by differentiated myotubes in another system we have also included differentiated C2C12. We now show that this mouse muscle cell line produces high amounts of BDNF after differentiation. In addition, C2C12 cells secrete BDNF following electrical stimulation further supporting the concept that myotubes secrete BDNF following activity (new Fig. 4). Most importantly, as suggested by the reviewer we have employed human primary myocytes and show that they also produce and secrete BDNF following differentiation (Fig. 6J-N). We are grateful to the reviewer for suggesting these important experiments that further strengthen the hypothesis that muscle-produced BDNF is important for β -cells activation in human as well.

2. A skeletal muscle immuno staining BDNF panel including Diaphragm and gastrocnemius muscle (or other muscle type used during exercise) is needed to show that 1) BDNF is expressed in the muscle and not in other cell types in diaphragm 2) BDNF is expressed in a muscle type that is directly affected by exercise.

Since, to our knowledge, there are no good antibodies against BDNF (all give a uniform staining and have not been validated in a BDNF KO mouse model) to perform immunohistochemistry experiments in muscle we have generated a new knock-in mouse model with a V5-tagged BDNF (Suppl. Fig 6 B-C). Immunostaining experiments in diaphragm and gastrocnemius muscles show very specific staining in fibers of both muscles (Fig. 4 A-F). Very interestingly, this experiment shows that not all fibers express BDNF, and even among the positive fibers some express BDNF at higher levels than others. We think that this new data, coupled with our new mouse model, will stimulate the muscle biology field to determine the identity of these fibers and the physiology behind the differential BDNF expression.

3. It would be warranted to show that BDNF infusion (in vivo) stimulates insulin secretion and whether BDNF exerts its effects in a dose-dependent fashion?

This is an important point in line also with the point 1 of reviewer 3, aimed at addressing circulating BDNF levels and their action on insulin secretion. We first would like to point out that the presence of BDNF in mouse plasma has been debated for years since nobody has been able to detect it by Western analysis. Most reports have relied on ELISA test results with questionable outcome. After extensive testing we have identified a very sensitive and specific BDNF antibody (Suppl. Fig. 6A) that has allowed us for the first time to show by western the presence of BDNF in plasma. However, its levels are very low (in the pg/mL range, Fig. 5G) and it can be detected only by I.P. and pooling plasma from different animals. After initial failed attempts to test if injection of BDNF changes insulin levels in vivo, guided by our experiments with isolated mouse islets (Fig. 2G-K), we used a different paradigm. We tested if BDNF infusion can change insulin levels in situation of hyperglycemia as found in the ex vivo studies with isolated mouse islets. We found that BDNF stimulated higher insulin levels when injected 10 minutes before glucose administration suggesting a direct link between circulating BDNF and insulin levels (Fig. 5I). Although at this point it is challenging to determine the dose-dependent effects of BDNF, the new finding that mice with muscle specific KO of BDNF have reduced circulating BDNF (Fig. 5G) and lower insulin levels during GTT further corroborate a direct relationship between plasma BDNF levels and insulin secretion, at least in situations of hyperglycemia.

4. I miss measurement of insulin sensitivity in the various mice models.

We apologize for the omission. We have now included ITT for the mouse models analyzed (New Fig. 2E, Suppl. Fig. 5C and Fig. 5F).

5. It appears that the muscle BDNF knockout mouse has a lower fasting glucose? Why?

We assume that the reviewer is referring to the lower insulin level in the fasting muscle BDNF KO mice (Fig. 5E). Indeed, we also noticed a reduction in the basal insulin levels but at this point, we do not know why. It is possible that BDNF may have some other peripheral functions, such as in regulating peripheral insulin receptor levels. However, the ITT test does not show any significant difference suggesting that this may be caused by a number of factors.

6. Finally, it would be interesting to do exercise studies (wheel running or treadmill running) in

the various mouse models and to study glucose and/or insulin levels to actually prove that BDNF is involved in a muscle-pancreatic endocrine loop.

The point is well taken, however, at this point we are unsure whether plasma BDNF is produced only during exercise. Indeed, on suggestion of the reviewer we have attempted some experiments with mice on short term exercise (1 hour) but have not detected any difference. After 10 days of exercise we see a trend but because of the very low plasma BDNF levels and the need to pool samples and concentrate by IP it is impossible to quantify. Although there is extensive literature on BDNF expression and exercise, it is still unclear what are the best exercise conditions to see a significant biological change. Also, as indicated above the very low levels of circulating BDNF (Suppl. Fig. 6A) makes this task challenging and future analysis with better methodologies may be needed to address this point.

Minor comments

The scales are different in figure 1D (Merge), most likely due to different islet sizes and that the picture have been adjusted. Please check the scales, write in the legend what the scale corresponds to if you don't add it to all the pictures. In picture Insulin/TrkB.T1-V5 merge, only part of the unit is visible. Please adjust.

We apologize for the oversight with the scale bars. We have now adjusted them and indicated that they correspond to 100 μm in all panels of Fig. 1D.

The specificity would have been even more convincing if you had added BDNF alone without glucose in Figure 2H (KO of the TrkB.T1 receptor).

To address this point we have performed new experiments and added the new data (Fig. 2K).

It would be good to have a y-axis for the calcium oscillations, as the authors have included in figure 5. It is difficult to compare the magnitude of the oscillations with other studies. It is not clear how you have performed the statistics in this analysis.

The point is well taken. We have now added the y-axis in all panels of Fig. 3 and quantification analysis for all panels with data on calcium oscillations, as suggested also by reviewer 2. We have also explained in the text how we did the statistics (Methods section page 18 lines 521-527)

The authors have included a useful qPCR panel over the expression of the different splice variants of TrkA, TrkB and TrkC in beta-tc-6 cells. However, when using UCSC in silico qPCR blast and NIH primer blast the used primers for TrkC.FL or TrkC.T1 do not have any template. That could explain why you apparently don't get any signal. Please correct.

On suggestion of the reviewer we have re-checked all primers and after blasting them against the NCBI mouse sequence database we confirmed that the primers included in the study are all correct. We are including below the sites that we have used for checking our primers. Nevertheless, we appreciate the opportunity to double check all primers used to avoid mistakes.

TrkC.FL primers are indicated below and generate an amplicon of 179 bp

http://genome.ucsc.edu/cgi-bin/hgc?o=78260298&g=htcUserAli&i=../trash/hgSs/hgSs_genome_4ff0a_d4d320.pslx+..%2Ftrash%2FhgSs%2FhgSs_genome_4ff0a_d4d320.fa+YourSeq&c=chr7&l=78260298&r=78356089&db=mm10&hgsid=779740959_PpSWmwa3Ae0pAB2iNnJvtR0nrfe

TrkC.T1 primers are indicated below and generate an amplicon of 186 bp

http://genome.ucsc.edu/cgi-bin/hgc?o=78304303&g=htcUserAli&i=../trash/hgSs/hgSs_genome_50cd2_d4dd60.pslx+..%2Ftrash%2FhgSs%2FhgSs_genome_50cd2_d4dd60.fa+YourSeq&c=chr7&l=78304303&r=78356089&db=mm10&hgsid=779741717_T10ew3oaG4On5CNAnaebjguaXkEF

In relation to that, you have included TrkB.T2 mouse primers in the primer table but do not include any TrkB.T2 measurements (and no specific product can be detected here either). Please only include what you have measured. NB: check supplementary
Also, please add the CT value range for TrkB.T1 for mouse and humans. It would be good to know the approximately expression levels.

The TrkB.T2 mouse primers were used for the expression analysis of TrkB isoform expression in mouse islets included in Suppl. Fig 1 of the original submission. The confusion may have come from the fact that we did not include TrkB.T2 expression analysis in B-TC6 cells. To make our analysis complete we have performed new RT-PCR analysis for TrkB.T2 and included the data in Fig. 5F. In addition, on suggestion of the reviewer we have added CT values for all mouse and human experiments (Suppl. Fig. 1; Fig 3F; Suppl. Fig. 7; Fig. 6P).

Reviewer #2 (Remarks to the Author):

Communication between insulin target tissues and beta-cells is at the heart of beta-cell compensation for insulin resistance. This paper proposes the novel idea that BDNF is secreted from muscle and then binds to the TrkB.T1 receptor in β -cells, leading to the stimulation of insulin secretion. Notably, knockout of BDNF expression in skeletal muscle, but not beta-cell produced glucose intolerance, pointing to skeletal muscle as the relevant source of the BDNF that regulates beta-cells. This is an important and novel finding.

We thank the reviewer for the positive comments and especially for noting the issues below.

1. The data presented in Figures 3H-3M are not accompanied by a quantitative data.

This is a valid point. We have now added quantification for all the calcium imaging analysis in Fig. 3.

2. The insulin responsiveness of the β -TC6 cell line is not shown; i.e. data comparable to that presented in Figs 2E-2H.

We thank the reviewer for this suggestion that makes the β -TC6 cell analysis more complete. The new data is included in the new Fig. 3H.

3. The second cell in Figure 5H appears not to be a beta-cell; in contrast to the first cell, Ca oscillations in this cell seems to fall off after treatment with BDNF.

The reviewer is correct and we apologize for the mistake. We have modified the figure to indicate the correct cell number 2 in Fig 6H, I.

Typos

1. Line 231 “following” should be “followed”
2. Line 272 “where” should be “were”
3. Line 291: “in obese diabetic mice model” should be “in an obese diabetic mouse model”

All typos have been corrected. Thank you.

Reviewer #3 (Remarks to the Author):

The manuscript by Fulgenzi et al. describes a novel mechanism by which the skeletal muscle regulates insulin secretion. The authors found that the truncated TrkB receptor (TrkB.T1) was expressed in beta cells of the pancreas in both mice and humans. They showed that activation of TrkB.T1 by BDNF enhanced glucose-induced insulin secretion in murine beta cells and directly stimulated insulin secretion in human beta cells. Removal of TrkB.T1 abolished the BDNF effect on insulin secretion. To support these in vitro findings, the authors further showed that mice lacking TrkB.T1 displayed impairments in glucose tolerance and glucose-induced insulin secretion. Additionally, the authors found that muscle-specific Bdnf knockout mice also showed impairments in glucose tolerance and glucose-induced insulin secretion. Therefore, the authors conclude that muscle-produced BDNF binds to TrkB.T1 in beta cells to promote insulin secretion. In general, the manuscript is well written, and the presented data are solid. The study not only uncovers a regulation mechanism of insulin secretion and a novel function for the truncated TrkB receptor, but also provide insights into the metabolic benefit of exercise. Thus, the manuscript is interesting to a broad readership. The following concerns should be addressed before the manuscript is published.

We appreciate the positive remarks about our study and the constructive feedback to strengthen our conclusions.

1. Additional evidence would be helpful to link muscle BDNF to TrkB.T1 in beta cells. BDNF produced in the muscle should reach the pancreas through the circulation. If the authors' model is correct, levels of BDNF in the circulation and the pancreas should be reduced in muscle-

specific Bdnf knockout mice. Alternatively, infusion of BDNF into the circulation should rescue the glucose-related phenotypes in the knockout mice.

These are indeed important experiments and we thank the reviewer for the suggestion. As indicated in the rebuttal to reviewer 1, the most important experiment to start addressing these points was the devising of a methodology to detect BDNF in the plasma (Suppl. Fig. 6A). This methodology that includes pooling of plasma from different animals and I.P. BDNF with a TrkB.Fc protein, has allowed us to show that muscle-specific BDNF KO mice have reduced circulating BDNF levels. Although this approach is not strictly quantitative, the fact that we obtained the same result at two different ages with pooled animals strongly support our conclusions (Fig. 5G). Secondly, using the same TrkB.Fc I.P. approach we have shown that BDNF is present in the pancreas and infusion of exogenous BDNF, not only increases BDNF pancreas levels (Fig. 5H) but also augments plasma insulin levels in response to glucose administration (Fig. 5I). Altogether, we think that these new data generated from the feedback of the reviewer further strengthen our hypothesis.

2. The authors point out that TrkB antibodies are not specific enough for immunohistochemistry. Instead, they used the antibodies for immunoelectron microscopy. The logic does not sound correct, because immune EM is one type of immunohistochemistry. The authors can perform in situ hybridization using ViewRNA or RNAscope in situ kits to show specific expression of TrkB.T1 in human beta cells.

The reviewer is correct and our explanation was not logical and modified the text. What we wanted to say is that the low levels of TrkB expression in islets made the co-staining with insulin, which is expressed at much higher levels, difficult. Instead, with the immune EM we could readily identify β -cells based on the presence of insulin in the cells expressing TrkB. Another important limitation was the limited number and size of the human islets that made it easier for us when handling for immune EM preparations. Nevertheless, prompted by the comment of the reviewer, closer observation of the immune EM data showed that the insulin granules were not very visible in Figure 6 G (See also point 4). Therefore, we have adjusted the magnification and included arrows pointing to the insulin granules. We also thank the reviewer for pointing out these new generation in situ hybridization kits which will be useful in future experiments.

3. Line 246: The statement “while spinal cord, pancreas and hypothalamus showed almost no recombination (Suppl. Fig. 6A)” is not accurate. Supplementary Fig. 6A shows significant recombination in the arcuate nucleus of the hypothalamus. The arcuate nucleus is an important brain site for the regulation of glucose homeostasis. To lessen the concern, the authors could cite published studies that show little or no expression of BDNF in this hypothalamic region.

The reviewer is correct since indeed there is some recombination in the arcuate nucleus of the hypothalamus. We thank the reviewer for pointing this out and the suggestion. We have cited published work to support the fact that there is little or no expression of BDNF in this region (lines 268-272).

4. The magnification in Figures 4I-L is too low to see insulin vesicles.

We are not sure if the reviewer meant the original Figure 2I-L or Figure 5F where indeed the power is too low to see the insulin vesicles. Nevertheless, to address this we have included the original full-resolution images from Fig 2 as separate files in the original data folder. In addition, we have increased the power and added arrows in the original Fig. 5F now Fig. 6G to clearly show insulin granules.

Reviewers' Comments:

Reviewer #1:

Remarks to the Author:

The authors did a great work and I have no further suggestions. I recommend acceptance of the manuscript.

Reviewer #2:

Remarks to the Author:

The authors have been very responsive to the reviewers' comments and have conducted new experiments when feasible, and corrected some of the issues related to the presentation of the data. This is an excellent paper.

Reviewer #3:

Remarks to the Author:

All of my concerns have been addressed in this revised manuscript. It is ready for publication.

I think there is a typo in the Figure 6G legend. b-cells are shaded in blue rather than in purple.

REVIEWERS' COMMENTS:

Reviewer #1 (Remarks to the Author):

The authors did a great work and I have no further suggestions. I recommend acceptance of the manuscript.

Reviewer #2 (Remarks to the Author):

The authors have been very responsive to the reviewers' comments and have conducted new experiments when feasible, and corrected some of the issues related to the presentation of the data. This is an excellent paper.

Reviewer #3 (Remarks to the Author):

All of my concerns have been addressed in this revised manuscript. It is ready for publication. I think there is a typo in the Figure 6G legend. b-cells are shaded in blue rather than in purple. We thank the reviewer for catching the typo that has now been fixed.